# Antibody-displaying extracellular vesicles for targeted cancer therapy

Oscar P. B. Wiklander [1,2,3,14] ✉, Doste R. Mamand[1,2,3,14], Dara K. Mohammad [4,5], Wenyi Zheng [1,3,6], Rim Jawad Wiklander[1,6], Taras Sych[7], Antje M. Zickler [1,3,6], Xiuming Liang[1,3,6], Heena Sharma[8], Andrea Lavado[8], Jeremy Bost[1,6], Samantha Roudi[1,3,6], Giulia Corso [1,6], Angus J. Lennaárd[1,6], Manuchehr Abedi-Valugerdi[1,6], Imre Mäger [9], Evren Alici[4,10], Erdinc Sezgin [7], Joel Z. Nordin [1,3,11], Dhanu Gupta[1,6,12], André Görgens [1,3,6,13] & Samir EL Andaloussi [1,3,6] ✉

Extracellular vesicles (EVs) function as natural delivery vectors and mediators of biological signals across tissues. Here, by leveraging these functionalities, we show that EVs decorated with an antibody-binding moiety specific for the fragment crystallizable (Fc) domain can be used as a modular delivery system for targeted cancer therapy. The Fc-EVs can be decorated with different types of immunoglobulin G antibody and thus be targeted to virtually any tissue of interest. Following optimization of the engineered EVs by screening Fc-binding and EV-sorting moieties, we show the targeting of EVs to cancer cells displaying the human epidermal receptor 2 or the programmed-death ligand 1, as well as lower tumour burden and extended survival of mice with subcutaneous melanoma tumours when systemically injected with EVs displaying an antibody for the programmed-death ligand 1 and loaded with the chemotherapeutic doxorubicin. EVs with Fc-binding domains may be adapted to display other Fc-fused proteins, bispecific antibodies and antibody–drug conjugates.

One of the major improvements in cancer therapy during the past decades pertains to targeted therapies with antibodies, such as anti-HER2 (human epidermal receptor 2) treatment against breast cancer and, more recently, immunotherapy with immune checkpoint inhibition[1]. One of the main targets is the programmed cell death protein 1 (PD-1) and its ligand PD-L1. Despite the overexpression of PD-L1 in various malignancies, only a subset of patients exhibit a durable response[2]. Strategies to improve the response rate include combinational approaches with chemotherapies, multiple checkpoint inhibition and other immune stimulatory approaches. These strategies are, however, limited in the coordinated delivery of the disparate therapeutics to the target of interest. An interesting alternative approach is the use of extracellular vesicles (EVs), which are promising nanocarriers for drug delivery. EVs are a heterogeneous group of natural nanovesicles that are secreted by all cells. These nanovesicles range in size from 30 nm to 2,000 nm in diameter and can impact neighbouring cells or cells

[1]Department of Laboratory Medicine, Unit for Biomolecular and Cellular Medicine, Karolinska Institutet, Stockholm, Sweden. [2]Breast Center, Karolinska Comprehensive Cancer Center, Karolinska University Hospital, Stockholm, Sweden. [3]Karolinska ATMP Center, ANA Futura, Huddinge, Sweden. [4]Department of Medicine Huddinge, Karolinska Institute, Stockholm, Sweden. [5]College of Agricultural Engineering Sciences, Salahaddin University-Erbil, Erbil, Iraq. [6]Department of Cellular Therapy and Allogeneic Stem Cell Transplantation (CAST), Karolinska University Hospital, Huddinge, Sweden. [7]Science for Life Laboratory, Department of Women's and Children's Health, Karolinska Institutet, Solna, Sweden. [8]Evox Therapeutics Limited, Oxford, UK. [9]Department of Physiology, Anatomy and Genetics, University of Oxford, Oxford, UK. [10]Hematology Center, Karolinska University Hospital, Stockholm, Sweden. [11]Department of Clinical Immunology and Transfusion Medicine (KITM), Karolinska University Hospital, Stockholm, Sweden. [12]Department of Paediatrics, University of Oxford, Oxford, UK. [13]Institute for Transfusion Medicine, University Hospital Essen, University of Duisburg-Essen, Duisburg, Germany. [14]These authors contributed equally: Oscar P. B. Wiklander, Doste R. Mamand. ✉e-mail: oscar.wiklander@ki.se; samir.el-andaloussi@ki.se

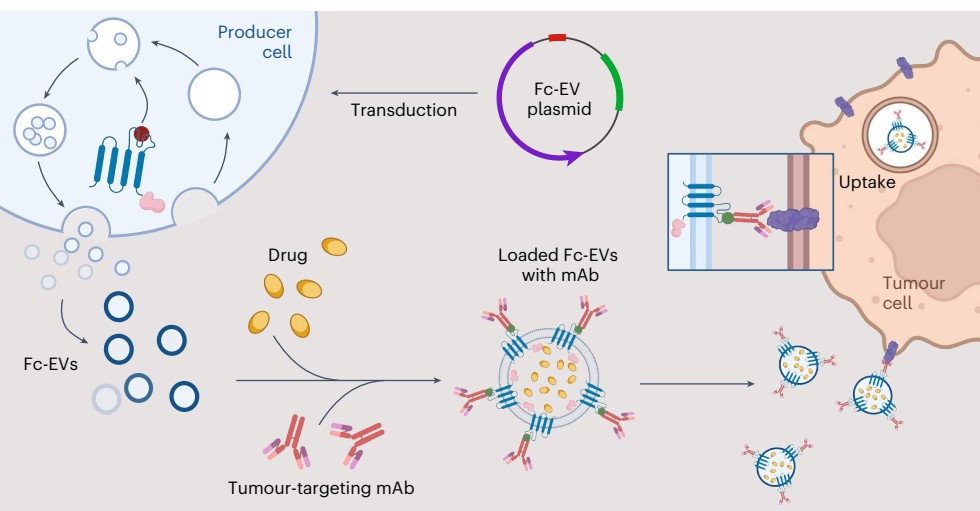

**Fig. 1 | Engineering cells to produce EVs decorated with an antibody-binding moiety specific for the Fc domain.** The producer cells are transduced with a construct to produce EVs that display an Fc-binding domain (Fc-EVs) that can be decorated with different types of antibody (mAb), loaded with therapeutic cargo (drug) and targeted to virtually any tissue of interest, such as cancer cells. Created with BioRender.com.

at a distance[3]. EVs contain lipids, proteins and nucleic acid species from the source cell and have the unique ability to convey these macromolecules via an advanced system of intercellular communication[4,5]. Importantly, EVs benefit from the ability to cross biological barriers to reach distant organs[6,7] and can be engineered to display targeting moieties and loaded with a wide variety of therapeutic cargo molecules[8,9]. In recent years, EVs have gained increasing attention, and there are currently numerous clinical trials being undertaken to evaluate the therapeutic potential of EVs[4].

In this Article, we propose a highly modular technology for EV therapeutics. Using molecular engineering tools, we have developed EVs that can bind the fragment crystallizable (Fc) portion of antibodies, so that the variable regions are displayed for antigen recognition. The Fc-binding EVs (Fc-EVs) can be decorated with different types of antibody and thus be targeted to essentially any tissues of interest. Here, the Fc-EVs technology is designed as a targeted cancer therapy using tumour-specific therapeutic antibodies to guide the EVs to tumour cells and to deliver antitumoural drugs.

## Results and discussion
### Optimization of Fc-EVs
The purpose of the Fc-EV technology was to combine the success of monoclonal antibodies for tissue targeting and the immense therapeutic potential of EVs for drug delivery[4,10]. Here, the aim was to develop a flexible system by displaying antibodies that can be interchanged depending on the intended target. To decorate the EVs with antibodies, an Fc-binding moiety was introduced to the EV surface. The EV-producing cells were engineered to express a fusion construct of an Fc-binding domain and an EV-sorting protein, to enable enrichment of the Fc domain on the EVs (Fig. 1). Given that there are many different available Fc-binding domains and various EV membrane-sorting domains, a systematic comparison of engineering strategies was conducted using similar strategies as recently reported by us[9]. The screening was conducted by producing fusion constructs combining each EV-sorting domain, which also expressed a fluorescent reporter, with each Fc-binding domain (Fig. 2a–c). The assessment was done by imaging flow cytometry (IFC), measuring the EV fluorescent reporter (mNG, mNeonGreen) and binding of fluorescently labelled (APC, allophycocyanin) immunoglobulin G (IgG) antibody (Supplementary Fig. 1a). The IFC used allowed detection of antibody binding to EVs at a single-vesicle level[11]. The initial assessment was conducted on the EV-producing cells (Supplementary Fig. 1b). First, the expression of

nine different EV-sorting domains, fused either C- or N-terminally to the reporter protein mNG, was assessed (Fig. 2a,b). The cells displayed the highest expression levels when the mNG was fused to the tetraspanins (cluster of differentiation 9 (CD9), CD63, CD81), annexin V or tumour necrosis factor receptor (TNFR) (Fig. 2b). The other EV-sorting domains showed lower mNG levels. Next, nine Fc binders (Fig. 2a) were chosen for the screen on cells to select the best candidates to bind antibodies using an APC⁺ IgG. Here, protein A and the derived z domains displayed higher antibody binding capacity compared to protein G and multidrug resistance protein 4, especially when fused to a tetraspanin or TNFR, whereas the other Fc displayed negligible binding (Supplementary Fig. 2).

Following the screen of cells, a comparison of engineered EVs with Fc binders was conducted. By comparing the conditioned media of the different EV-producing cells expressing the nine different EV-sorting domains, the TNFR and tetraspanins, especially CD63 C-terminal fusions, outcompeted the other EV-sorting domains in terms of mNG-positive (mNG⁺) events per millilitre, indicating a greater production of engineered EVs (Fig. 2c). The EVs were thus designed to display either protein A, z domains or 4z domains, by fusion with the tetraspanins or TNFR. All constructs generated engineered EVs with Fc binders, and the combination of CD63–z generated the highest level of expression (Fig. 2d). All EVs showed antibody binding capacity, with similar high binding efficiency among the tetraspanin constructs when using the z domain, but it was significantly lower using TNFR as an expression scaffold (Fig. 2d). A similar tendency was observed when using 4z or protein A as Fc binder. As CD63–z fusions also resulted in engineered EVs with the highest level of expression (Fig. 2e), the CD63–z domain combination was selected as the best candidate and was thus used in the subsequent experiments, henceforth denoted Fc-EVs.

### Characterization of Fc-EVs and confirmation of specific antibody binding
There is an intrinsic complexity of characterizing EVs, and several complementary characterization methods are thus needed[12]. When analysed by nanoparticle tracking analysis (NTA), the Fc-EVs displayed a peak size of approximately 100 nm, which is within the range of small EVs (Fig. 3a). The size of approximately 100 nm was further confirmed by immune electron microscopy, which showed binding of nanogold particle-labelled antibodies to Fc-EVs but not to the Fc-negative control EVs (ctrl-EVs) (Supplementary Fig. 3a). To further confirm the binding of antibody to Fc-EVs, size exclusion chromatography (SEC)[13–15] was conducted following incubation of mNG-labelled Fc-EVs or ctrl-EVs with

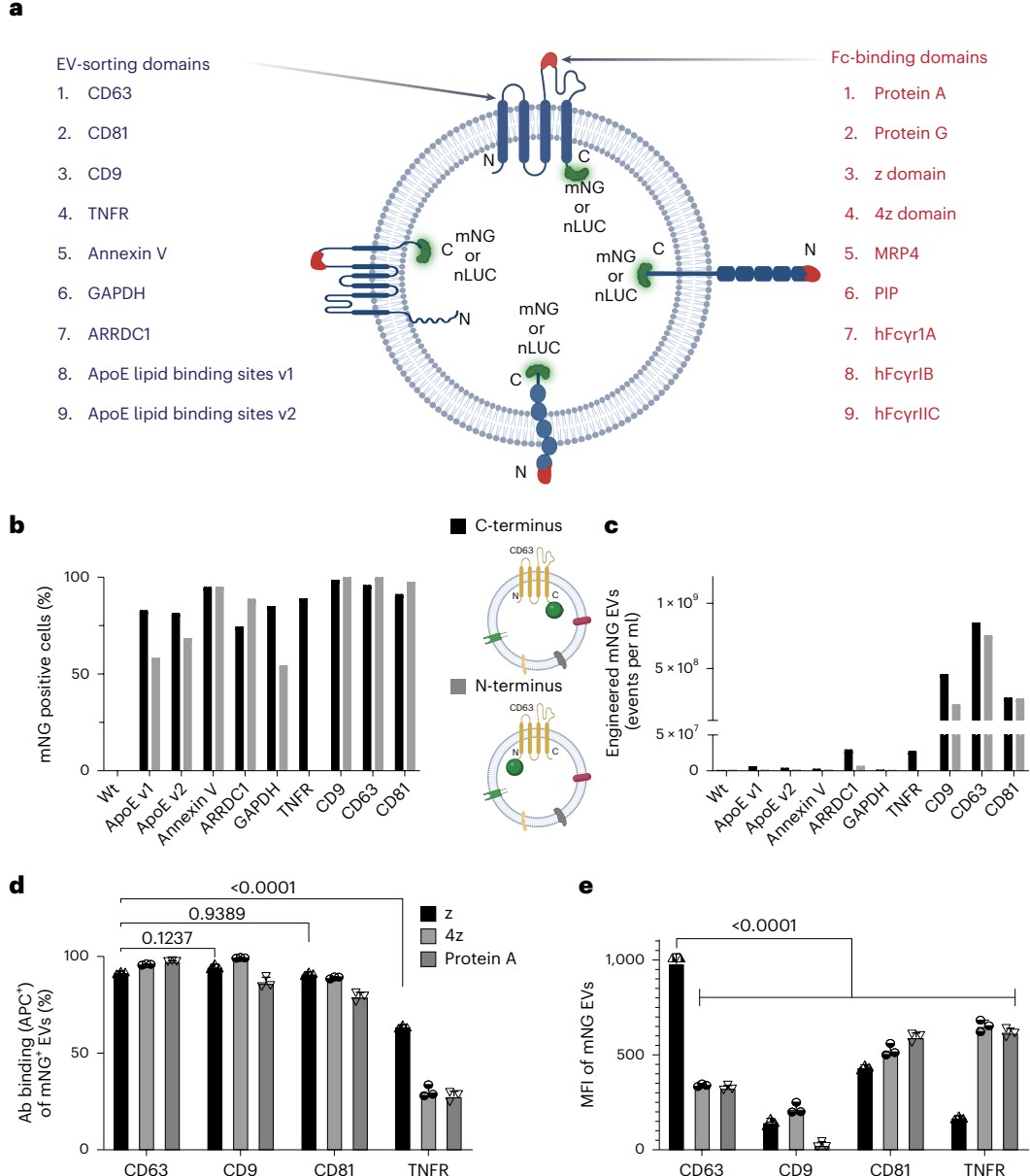

**Fig. 2 | Screening of EV-sorting proteins and Fc-binding domains. a**, Nine EV-sorting domains and nine Fc-binding domains used in different combinations, as illustrated by the examples in the centre, for the screen to optimize Fc-EVs. **b**, IFC of transient transfection of HEK293T cells with nine different EVs sorting domains, intraluminally fused via the N- or C-terminal to the fluorescent protein mNG (TNFR only have one intraluminal terminus, C-terminus). Wt, wild type. **c**, IFC comparison of engineered mNG EVs from cells in both configurations based on events in conditioned media. **d**, IFC analysis of the expression of the Fc binder based on the mean fluorescence intensity (MFI) of mNG for the top candidate combinations of different Fc binders (z, 4z, protein A) with different EVs sorting domains (CD63, CD9, CD81, TNFR). **e**, IFC analysis of the expression of the Fc binder based on the percentage of double positive (APC Abs binding

mNG-Fc-EVs) for the top candidate combinations of different Fc binders (z, 4z, protein A) with different EV-sorting domains (CD63, CD9, CD81, TNFR). Other combinations can be found in Supplementary Fig. 2. Illustrations created with BioRender.com. GAPDH, first 148 amino acids (aa) of glyceraldehyde-3-phosphate dehydrogenase; ARRDC1, arrestin domain-containing protein; ApoE lipid binding sites (v1 corresponds to 355 aa, v2 corresponds to 1,009 aa); z domain, synthetic Fc region-binding domain originated from the B domain of protein A[50]; 4z, four repetitive z domains; MRP4, multidrug resistance protein 4; PIP, prolactin-induced protein; hFcγr, human Fc gamma receptor. Panels **d** and **e** are shown as mean ± s.d. $n = 3$ biological replicates. Statistical significance was calculated using two-way ANOVA with Dunnett's post-test compared with each value; $P$ values are indicated above the plots of **d** and **e**.

APC⁺ IgG. The majority of mNG⁺ Fc-EVs and ctrl-EVs was observed in the expected fractions 3–6 as detected by mNG fluorescence (Fig. 3b, left $y$-axis). However, the APC antibodies co-localized only in the same fractions when incubated with Fc-EVs but not ctrl-EVs (Fig. 3b, right $y$-axis), confirming Fc-EV association with antibody. Western blot was implemented on the Fc-EVs, which confirmed the presence of the classical EV markers TSG101, CD63[12] and the CD63-fused luminescent reporter nano-luciferase (nLuc, used as Fc-EV reporter for in vivo testing below),

which confirms successful engineering (Supplementary Fig. 3b). The Fc-EVs were subsequently analysed using bead-based multiplex flow cytometry[16,17]. Following incubation of fluorescently labelled human Fc fragments with Fc-EVs, a clear shift in fluorescence intensity was observed for all beads, whereas this shift was not seen with ctrl-EVs or without EVs (Supplementary Fig. 3c), thus indicating efficient binding of Fc fragments and capture beads to Fc-EVs. Next, IFC was again used to further assess antibody binding to EVs. Fc-negative ctrl-EVs did not

bind the APC⁺ IgG, whereas Fc-EVs showed a clear dose-dependent binding when incubated with increasing IgG amounts (Fig. 3c and Supplementary Fig. 3d,e). IFC was further used throughout the study to control for the antibody-binding capacity of the Fc-EVs. Next, the affinity of different IgG subtypes to Fc-EVs was assessed. In line with previous reports of protein A and z domain[18,19], human IgG1 (hIgG1) had the greatest affinity to Fc-EVs as assessed by incubating increasing doses of different IgG subtypes with the EVs (Fig. 3d). This is also an important consideration as most clinically approved therapeutic Abs are hIgG1 based[20].

In addition, to assess whether Fc-EVs would retain the displayed antibodies when exposed to blood, a stability assay was performed with human or mouse plasma. mNG⁺ Fc-EVs were incubated with APC-labelled IgG and then exposed to either phosphate-buffered saline (PBS) or human or mouse plasma, with no loss of APC labelling among mNG⁺ Fc-EVs as shown by unchanged double positivity (of mNG and APC) in IFC, at 1 to 30 min exposure (Supplementary Fig. 3f). Similarly, the double positivity was unaffected by additional incubation with different non-APC-labelled IgG subtypes (Supplementary Fig. 3g). If needed, alternative approaches to further functionalize the antibody display could have been considered, as previously explored by others utilizing recombinant phosphatidylserine binding of nanobodies[21] or more recently by EV display of avidin that can be combined with biotinylated antibodies[22]. However, as the Fc-EVs showed robust antibody display, further functionalization was not required for the current study. Moreover, cellular uptake of APC⁺ IgG co-localized with mNG⁺ Fc-EVs in HeLa cells, whereas no antibody could be detected when cells were treated with naked IgG or in combination with mNG⁺ ctrl-EVs (Supplementary Fig. 3h). Furthermore, imaging by nanoimager[23], was utilized to further assess the association of Fc-EV and antibody at a detailed level. Fractions from the SEC of Fc-EVs with or without antibodies confirmed the association of green Fc-EVs and red antibody, rendering yellow particulates, (Fig. 3e,f). Finally, to enumerate the number of antibodies displayed per Fc-EV, single-particle profiler (SPP) was used. SPP is a recent method based on fluorescence fluctuations, which has been developed for biophysical profiling and measurement of nano-sized bioparticles[24]. Using this technique, mNG⁺ Fc-EVs were shown to display an average of 105 APC-labelled hIgG per vesicle, whereas Fc-EVs did not display mouse IgG (≤1 Ab/EV), and ctrl-EVs with human or mouse IgG all presented negligible amounts of IgG with ≤1 Ab/EV (Fig. 3g–i and Supplementary Fig. 3i–k). Taken together, the different characterization techniques all showed efficient and stable binding of antibodies to Fc-EVs, which encouraged us to further investigate targeted delivery of Fc-EVs guided by antibodies in vitro and in vivo.

### Antibody-guided Fc-EVs as a versatile targeting technology

To demonstrate the multifaceted potential of the Fc-EV technology, two different relevant therapeutic targets were assessed in vitro. HER2 is an oncogene that is amplified and used as a biomarker and therapeutic target in several cancer types[25]. Here, the clinically used HER2 antibody trastuzumab was incubated with Fc-EVs and assessed in HER2 positive breast cancer cells (SKBR-3). A 339-fold increase of EV uptake, as measured by mean fluorescent intensity (MFI), was observed in the cells when the mNG⁺ Fc-EVs were guided by trastuzumab compared to Fc-EVs without antibody (Fig. 4a). Ctrl-EVs displayed no increased

uptake with any of the antibodies, and the Fc-EV uptake was unaffected by control-Ab. As expected, pre-treatment of the SKBR-3 cells with naked trastuzumab significantly decreased the uptake of Fc-EVs displaying trastuzumab (Supplementary Fig. 4a). However, the decrease was quite modest, most probably due to the rapid recycling of HER2[26]. Taken together, this shows that the system is highly specific and that the antibody indeed drives the enhanced uptake.

Next, the common target for immune checkpoint inhibition, PD-L1, which is overexpressed in many cancer types and binds the immune inhibitory molecule PD-1, was investigated as a target for Fc-EV delivery. A comparison of different PD-L1 antibodies (PD-L1-Ab) showed that the clinically used atezolizumab resulted in the greatest increase in EV uptake in the malignant melanoma cells (B16F10) and was used as the PD-L1-Ab for subsequent experiments (Supplementary Fig. 4b). This cell line is commonly used as an in vivo melanoma tumour model and is known to overexpress PD-L1 in vivo[27]. To increase the expression levels of PD-L1 in vitro and thus mimic the levels of the tumour in vivo, interferon gamma (IFNγ) can be used to stimulate the cells[28]. As shown in Fig. 4b, Fc-EVs displayed a 509-fold increase in uptake in IFNγ-stimulated B16F10 cells when guided by atezolizumab compared to Fc-EVs without antibody. The increased uptake of Fc-EVs displaying PD-L1-Ab was further confirmed by microscopy analysis of treated B16F10 cells, where the Fc-EVs could be detected only when displaying PD-L1-Ab (anti PD-L1 APC and atezolizumab) (Fig. 4c and Supplementary Fig. 4c).

Furthermore, the proportion of cells taking up mNG⁺ Fc-EVs dramatically increased by display of targeting antibody. In HER2⁺ SKBR-3 cells, display of HER2-Ab increased the percentage of mNG⁺ cells from 7% to 82% and in PD-L1⁺ B16F10 cells, display of PD-L1-Ab led to an increase from 11% to 76% (Supplementary Fig. 4d,e). As hypothesized, display of PD-L1-Ab on Fc-EVs had no effect on uptake in PD-L1-negative SKBR-3 cells, and HER2-Ab had no effect in HER2⁻ B16F10 cells (Supplementary Fig. 4d,e), which further underpins the target specificity of Fc-EVs. Taken together, these findings clearly show that Fc-EVs can be specifically targeted to cells by antibodies and thus motivated further investigation of the targeting potential in vivo.

### Antibody-guided Fc-EVs target tumours in vivo

Fc-EVs were next assessed in B16F10 tumour-bearing mice by intravenous (IV) injection of nLuc⁺ Fc-EVs alone (Fc-EV) or combined with PD-L1-Ab (atezolizumab, Fc-EV + PD-L1-Ab) or isotype control (Fc-EV + IgG-ctrl) (Fig. 5a). Owing to the nLuc reporter of the Fc-EVs, the amount of EVs in tissue lysates could be determined based on luminescence measurements.

Display of PD-L1-Ab significantly increased the accumulation of Fc-EVs into tumour tissue over time compared to Fc-EVs alone, whereas the isotype control did not affect the Fc-EV accumulation in tumours (Fig. 5b). The tumour accumulation reached the highest levels at 30 min post injection and displayed a declining yet significantly increased accumulation until 72 h after injection in mice treated with Fc-EV + PDL1-Ab. This treatment group displayed a significantly increased tumour accumulation at each time point from 15 min to 72 h, ranging from 6- to 180-fold more than Fc-EVs alone, whereas the isotype control did not affect the Fc-EV accumulation in tumours (Fig. 5c and Supplementary Fig. 6a–f). This corresponds to a tumour accumulation

**Fig. 3 | Fc-EV characterization. a**, NTA showing similar size distribution of Fc-EVs and control (ctrl-EVs) with and without antibody (REA(S)-APC, human IgG1, Ab). **b**, Fluorescence reading of 0.5 ml fractions from SEC of mNG representing EV, and APC representing Ab. Upper panel: association of Fc-EVs and Ab at EV expected fractions 3–6; lower panel: no association in the EV fractions between ctrl-EVs and Ab. a.u., arbitrary units. **c**, IFC data including example event images that show binding of APC-labelled human IgG isotype control antibodies to mNG-Fc-EVs but not to mNG-ctrl-EVs. **d**, Binding affinity of IgG subtypes to Fc-EVs based on binding of mNG⁺ Fc-EVs with PE⁺ IgG-subtypes (expressed as percentage

of double positive (DP) (PE and mNG) over mNG). APC mean fluorescence intensities following staining of Fc-EVs with increasing doses of human isotype control IgG antibodies. **e**, Fluorescence reading of fractions from SEC. Upper panel: Fc-EVs alone; lower panel: association of Fc-EVs and Ab at EV expected fractions. **f**, Nanoimager microscopy images of selected fractions from SEC (**e**), showing green fluorescent EVs (without detectable red antibody) only for the ctrl-EVs, but double positivity (yellow) for the Fc-EVs and Ab. **g–i**, Quantification of antibodies per EVs was performed using the SPP. Histogram of number of hIgG to Fc-EVs (**g**), number of mIgG to Fc-EVs (**h**) and number of hIgG to ctrl-EVs (**i**).

of $7.25 \times 10^9$ EVs or 7.25% of the injected dose (normalized to gram tissue) at 30 min for Fc-EV + PD-L1-Ab (Fig. 5b and Supplementary Fig. 6g). In line with previous findings[29,30], the half-life in plasma of Fc-EVs was 3–4 min with no significant difference between Fc-EVs alone or when displaying PD-L1-Ab or IgG-ctrl (Fig. 5d).

In parallel, regional lymph nodes had an increased uptake of Fc-EVs when PD-L1-Ab was displayed (Supplementary Fig. 7a). This is in line with the findings that tumour-educated immune cells in lymph nodes upregulate PD-L1, which restrains tumour-specific immunity and appears to be an important cancer therapy target[31]. As

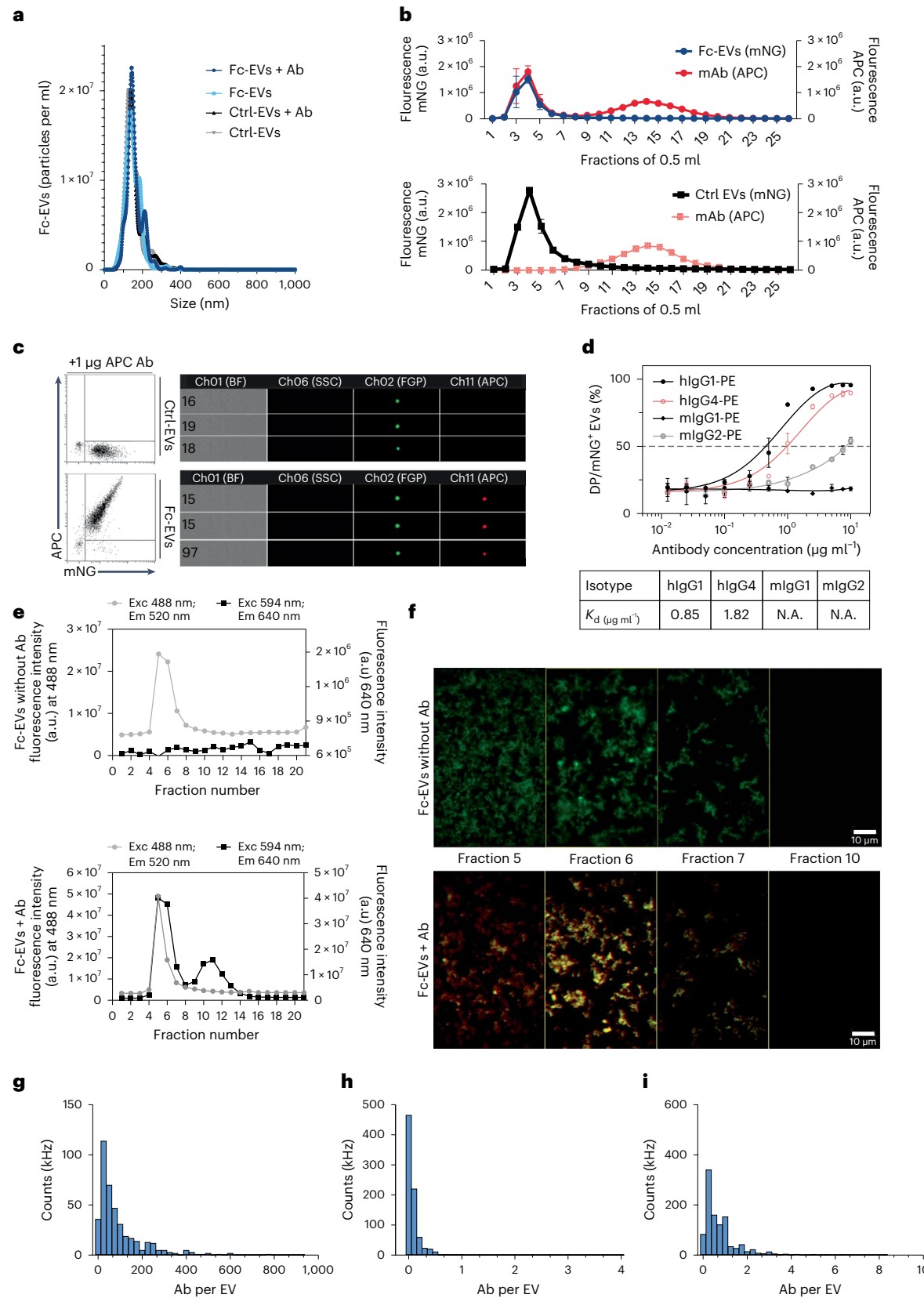

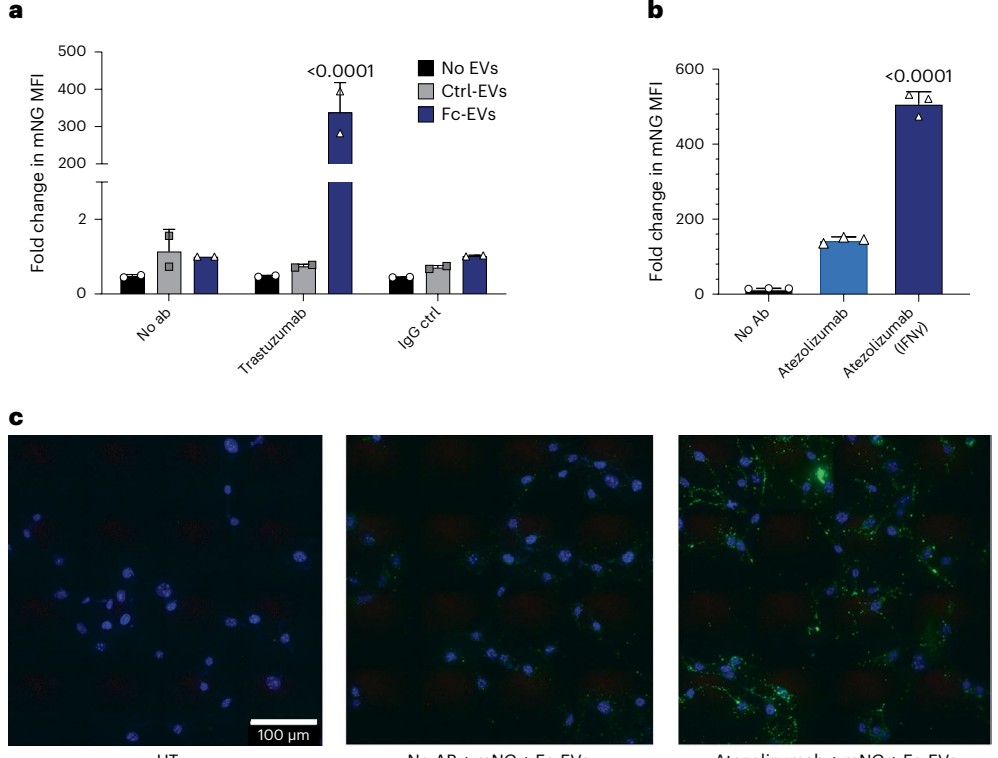

**Fig. 4 | Fc-EV targeting using antibodies. a**, Flow cytometry measurement of uptake by MFI of mNG⁺ EVs (Fc-EVs versus ctrl-EVs versus no EVs) in HER2 positive breast cancer cells (SKBR-3 cells) when incubated with HER2-Ab (trastuzumab), control-Ab (IgG ctrl) or no Ab, showing significant increase in uptake of Fc-EVs by trastuzumab. **b**, Flow cytometry measurement of uptake (by MFI) of mNG⁺ Fc-EVs decorated with the PD-L1-Ab atezolizumab in PD-L1 expression stimulated (IFNγ) or unstimulated malignant melanoma (B16F10) cells. **c**, Fluorescence microscopy images of B16F10 cells stained with DAPI (blue), untreated (UT) or treated with mNG⁺ (green) Fc-EVs alone or with the PD-L1-Ab atezolizumab, showing increased uptake of the Fc-EVs when decorated with PD-L1-Ab. Panels **a** and **b** are shown as mean ± s.d. *n* = 3 biological replicates. Statistical significance was calculated using one-way (**b**) or two-way (**a**) ANOVA with Tukey's post-test compared with each value; *P* values are indicated above the plots throughout.

expected, slightly lower levels of EVs were detected in spleen and liver for Fc-EV + PD-L1-Ab compared to Fc-EV + IgG-ctrl. These differences were, however, only significant at later time points of spleen accumulation (Fig. 5e,f). The altered tissue distribution further emphasizes the targeting efficiency with enrichment in PD-L1-positive tissue and a seemingly reduced uptake of Fc-EV + PD-L1-Ab in spleen and liver. Furthermore, liver and spleen are organs of the mononuclear phagocyte system, known for clearing untargeted nanoparticles, which is considered one of the hurdles in nanomedicine[32]. The sustained increased accumulation is, in addition to clearly showing tumour targeting properties, also in line with the enhanced permeability and retention effect associated with nanoparticles[33]. These findings in combination with the reduced mononuclear phagocyte system uptake thus further underscore the targeting of the PD-L1-Ab displaying Fc-EVs as a potential cancer therapy.

Next, an additional experiment was conducted in B16F10 tumour-bearing mice with the aim to investigate the involvement of specific immune cell types in the PD-L1 Fc-EV targeted technology. The set-up was as depicted in Fig. 5a, but with the injection of mNG⁺ Fc-EVs, allowing for fluorescence-based detection using flow cytometry of single-cell suspensions from isolated tissues (Supplementary Fig. 5). Consistent with the tissue distribution findings above, Fc-EVs were enriched in tumour cells when displaying PD-L1-Ab compared to control antibody, especially in PD-L1⁺ tumour cells (Fig. 5g,h and Supplementary Fig. 7b). Next, the tumour tissue immune cells were assessed. The majority of the tumour immune cells were CD3⁺ T cells (Supplementary Fig. 7c). Ninety-nine per cent of the T cells were CD3⁺CD8⁻ (Supplementary Fig. 7d), and 95% of these were found to

be PD-L1⁺ (Supplementary Fig. 7e). It is noteworthy that PD-L1⁺ T cells have been shown to have diverse tolerogenic effects on tumour immunity and are thus a potential important target of cancer therapies[34]. Interestingly, and as expected, Fc-EVs were significantly enriched in these PD-L1⁺ T cells when guided by PD-L1-Ab (Fig. 5i). No difference in Fc-EV uptake was seen in B cells, monocytes and macrophages, which also displayed a lower degree of PD-L1 positivity compared to T cells (Supplementary Fig. 7e–h).

Furthermore, to assess the broader utility of Fc-EVs for tumour targeting, a xenograft mouse model based on human HER2⁺ breast cancer was used by inoculating SKBR-3 cells in the mammary gland[35]. Following tumour formation, nLuc⁺ Fc-EVs displaying HER2-Ab (trastuzumab, Fc-EV + HER2-Ab) or isotype control (Fc-EV + IgG-ctrl) were injected intravenously (Fig. 6a). Similar to the PD-L1 targeting in the B16F10 tumour model above (Fig. 5b), display of HER2-Ab led to an increased tumour accumulation in SKBR-3 tumour-bearing mice (Fig. 6b). In absolute numbers, the Fc-EV + HER2-Ab tumour accumulation was 4.6 × 10⁵ (0.0002% of injected dose) (Supplementary Fig. 8a,b), which is lower than for PD-L1 targeting in B16F10 (Supplementary Fig. 6g). This difference can be explained by the relatively smaller tumours (about 0.05 g) of the SKBR-3-bearing mice (Supplementary Fig. 8c) compared to B16F10-bearing mice that had an average tumour weight of about 0.3 g (Supplementary Fig. 6h). Furthermore, only 15% of the cells in the dissected SKBR-3 tumours were HER2⁺ (Supplementary Fig. 8d) compared to 50% of the cells being PD-L1⁺ in the dissected B16F10 tumours (Supplementary Fig. 7b). As expected in a xenograft model with no human HER2⁺ cells present in normal tissues, no significant differences was seen in terms of spleen and liver distribution (Fig. 6c,d).

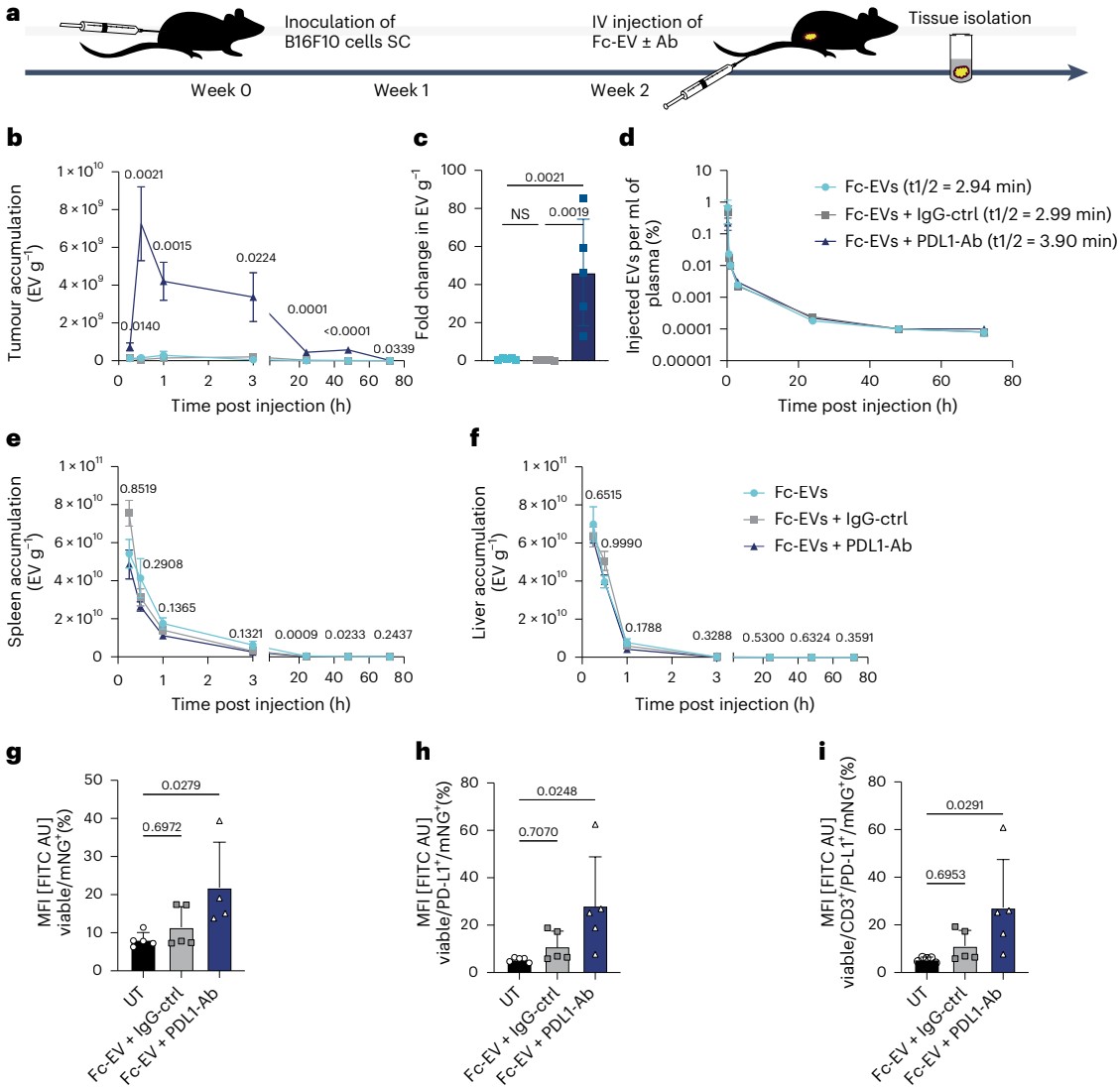

**Fig. 5 | Fc-EV tumour targeting using PD-L1-Ab.** IV injection of Fc-EVs with PD-L1-Ab (atezolizumab, Fc-EV + PD-L1-Ab), control-Ab (Fc-EV + IgG-ctrl) or no Ab (Fc-EV) in malignant melanoma (B16F10) tumour-bearing mice. **a**, The experimental set-up with inoculation of B16F10 cells followed by tumour formation for 2 weeks before IV injection of Fc-EVs (nLuc⁺ Fc-EVs for **b**–**f**, mNG⁺ Fc-EVs for **g**–**i**) with or without Ab, followed by tissue or blood collection. SC, subcutaneously. **b**, Detected EVs (based on luminescence) per gram tumour tissue at indicated time points post injection. **c**, Fold change in detected EVs in tumour at 30 min post injection. **d**, Percentage of injected EV dose found in plasma at different time points with indicated half-life based on one phase decay.

**e**,**f**, Detected EVs per gram spleen (**e**) or liver (**f**), at indicated time points post injection. **g**–**i**, Flow cytometry analysis of single-cell suspensions of tumour tissue following IV injection of mNG⁺ Fc-EV with PD-L1-Ab or control-Ab. **g**, Percentage of tumour cells that have taken up mNG⁺ Fc-EVs, based on viable cells from single-cell suspension of the tumour that are mNG⁺. **h**, Percentage of PD-L1⁺ tumour cells that have taken up mNG⁺ Fc-EVs. **i**, Percentage of PD-L1⁺ T cells (CD3⁺) in tumour tissue that have taken up mNG⁺ Fc-EVs. All data are shown as mean ± s.d. *n* = 5 biological replicates. Statistical significance was calculated using one-way ANOVA with Tukey's (**b**–**f**) or Dunnett's (**g**–**i**) post-test compared with each value; *P* values are indicated above the plots throughout.

## PD-L1-Ab displaying cytotoxic Fc-EVs as therapy for malignant melanoma

Given the successful tumour targeting of Fc-EVs with the PD-L1-Ab, we set out to investigate whether Fc-EVs could be utilized as a delivery vehicle for targeted delivery of chemotherapeutic drug. The hypothesis was that the PD-L1 targeting approach would result in a dual therapeutic role as it both directs the drug-loaded Fc-EVs to the tumour and the PD-L1-Ab itself functions as a therapeutic intervention by blocking the immunosuppressive PD1/PD-L1 axis. Importantly, although effective in vitro, chemotherapeutic drugs have no or very little effect in melanoma animal models and patients. The lack of effect is multifactorial and includes the fact that the doses needed for therapeutic effect are greater than the doses causing intolerable adverse events such as cardiac toxicity[36]. The treatment with the checkpoint inhibitors is also associated with severe adverse events[37,38]. There is thus a great need

to improve the drug potency and decrease the systemic side effects, which might be overcome with targeted therapies. In parallel, we have previously published an optimized protocol for electroporation of the chemotherapy doxorubicin (Dox) into EVs. The loading technique showed robust loading and recovery of EVs, as well as improved drug potency, with a 190-fold increased cytotoxicity on B16F10 cells compared to naked Dox[39]. Here, electroporation of Fc-EVs yielded a dose of 0.34 µg Dox per 1 × 10¹¹ EVs (Supplementary Fig. 9a). Dox-loaded Fc-EVs had a significant cytotoxicity on B16F10 cells irrespective of PD-L1-Ab compared with untreated cells (Supplementary Fig. 9c). Without Dox present, Fc-EVs and PD-L1-Ab, by themselves or in combination, had no effect. As electroporation causes permeabilization of the membrane, the stability of encapsulation of Dox was assessed in plasma at 37 °C (Supplementary Fig. 9b). Only a slight decrease of 20% was observed at 30 min. Given that EVs rapidly distribute to their target tissue within

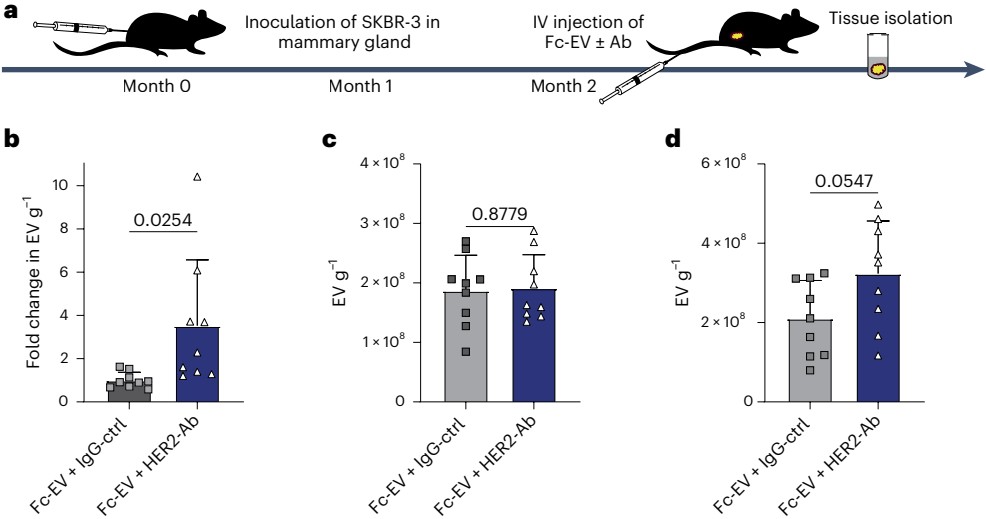

**Fig. 6 | Fc-EV tumour targeting using HER2-Ab.** IV injection of Fc-EVs with HER2-Ab (trastuzumab, Fc-EV + HER2-Ab) compared to control-Ab (Fc-EV + IgG-ctrl) in HER2⁺ breast cancer (SKBR-3) tumour-bearing Swiss nude mice. **a**, The experimental set-up with inoculation of SKBR-3 cells followed by tumour formation for 2 months before IV injection of nLuc⁺ Fc-EVs with Abs, followed by tissue collection 30 min post injection. **b**, Fold change in detected EVs (based on luminescence) per gram tumour tissue compared to Fc-EV + IgG-ctrl. **c**,**d**, Accumulation of Fc-EV + HER2-Ab (based on luminescence) compared to Fc-EV + IgG-ctrl per gram spleen (**c**) and liver (**d**). All data are shown as mean ± s.d. *n* = 10 mice. Statistical significance was calculated using two-tailed unpaired *t*-test analysis compared with each value; *P* values are indicated above the plots throughout.

minutes[29,30], and the relatively low dose of Dox required when encapsulated in EVs[39], PD-L1 targeted Dox-loaded Fc-EVs were considered a highly attractive candidate for proof-of-concept use of the Fc-EV system in cancer settings.

To investigate whether the Fc-EV technology could be used as cancer therapy with low therapeutic doses, B16F10 melanoma-bearing mice were systemically treated every third day with Fc-EVs loaded with Dox and decorated with PD-L1-Ab (Fig. 7a). As control, PBS, single-entity therapy with 25 µg PD-L1-Ab or control antibody, 1 × 10¹¹ Fc-EV or 50 µg Dox was compared to 1 × 10¹¹ Fc-EVs loaded with 0.34 µg Dox and decorated with 25 µg PD-L1-Ab, control antibody or no antibody (Fc-EV + Dox + PD-L1-Ab, Fc-EV + Dox + ctrl-Ab, Fc-EV + Dox, respectively) or 1 × 10¹¹ Fc-EVs without Dox decorated with PD-L1-Ab (Fc-EV + PD-L1-Ab). Single-entity therapy did not significantly affect the disease course in terms of tumour progression or survival (Supplementary Fig. 10a–c). However, Fc-EV + Dox + PD-L1-Ab significantly suppressed tumour progression with decreased tumour size development (Fig. 7b) and significantly improved survival (Fig. 7c). In fact, this combinational therapy displayed 100% survival at the measured end point of 20 days, compared to 35% survival in mock treated mice.

These highly encouraging data motivated us to evaluate further whether this therapy would be effective over a longer period as well, and a second experiment was conducted with a pre-set end point of 35 days (Fig. 7d). As expected, Fc-EV + Dox + PD-L1-Ab treatment again showed a significantly improved disease course with decreased tumour size development over time (Fig. 7e) and a significantly improved survival (Fig. 7f). It is noteworthy that the median survival of mock treated mice was 18 days, with 24.5 days for Fc-EV + Dox decorated with control antibody, whereas median survival had not yet been reached for Fc-EVs + Dox + PD-L1-Ab. The treatments did not give rise to any observable symptoms, and measurement of kidney and liver function did not reveal any differences compared to mock treated control (Supplementary Fig. 11a–d).

## Outlook

EVs are increasingly being investigated as promising biotherapeutics owing to their potential as a natural delivery vector that can transport macromolecular content and surpass biological barriers to the site of interest. Compared to other vehicles, such as liposomes and polymer-based synthetic nanoparticles, EVs are associated with high biocompatibility, minimal toxicity and enhanced drug potency, with improved pharmacokinetic profiles including improved tumour penetrance and retention in tumour cells[39–43]. In comparison to synthetic nanoparticles, the EV field is, however, still young, and despite the promising delivery profile and low toxicity in early-phase clinical trials, caution and more data are still needed[44]. Using EVs as nanocarriers further provides the possibility to specifically bioengineer them to express targeting moieties, which we and others have reported previously[4,45]. Here, the aim was to develop a versatile system by displaying antibodies. This offers an advantage over conventional EV targeting systems expressing a prespecified targeting moiety as the antibodies here can be interchanged depending on the intended target, thereby offering the potential to utilize Fc-EVs as an off-the-shelf modular drug that can be combined with previously approved therapeutic antibodies.

Others have very recently shown that the concept of displaying antibodies on EVs is feasible in vitro[22,46]. The targeting capacity of these EVs were, however, modest with less than 1.5-fold increased uptake of the engineered EVs in the target cell[46]. We optimized the antibody display on EVs through a screen of nine different EV-sorting domains and nine Fc-binding domains and characterized and demonstrated that Fc-EVs can be used to enrich EVs by several hundredfold to a cellular target, such as the therapeutically relevant HER2 and PD-L1. As proof of concept, we further demonstrated that PD-L1-Ab displaying EVs show a sustained and significantly enriched accumulation in tumour tissue following systemic injection. Furthermore, when loaded with chemotherapeutic drugs, PD-L1-Ab displaying EVs led to an improved anti-tumoural effect of the drug. Improving the cytotoxic efficiency and targeting delivery of Dox could allow for prolonged effective use of Dox, which is otherwise limited by accumulative cardiac toxicity.

In conclusion, the Fc-EV technology offers a combined EV-antibody therapy in which targeting and/or therapeutic antibodies facilitate targeted EV delivery with therapeutic cargo that can generate a synergistic effect. In theory, the technology could be applied to a myriad of indications and used in multiple combinations, not only limited to classical antibody but could also including Fc-fused proteins, antibody–drug conjugates and bi-specific antibodies, to mention a few. Subsequent

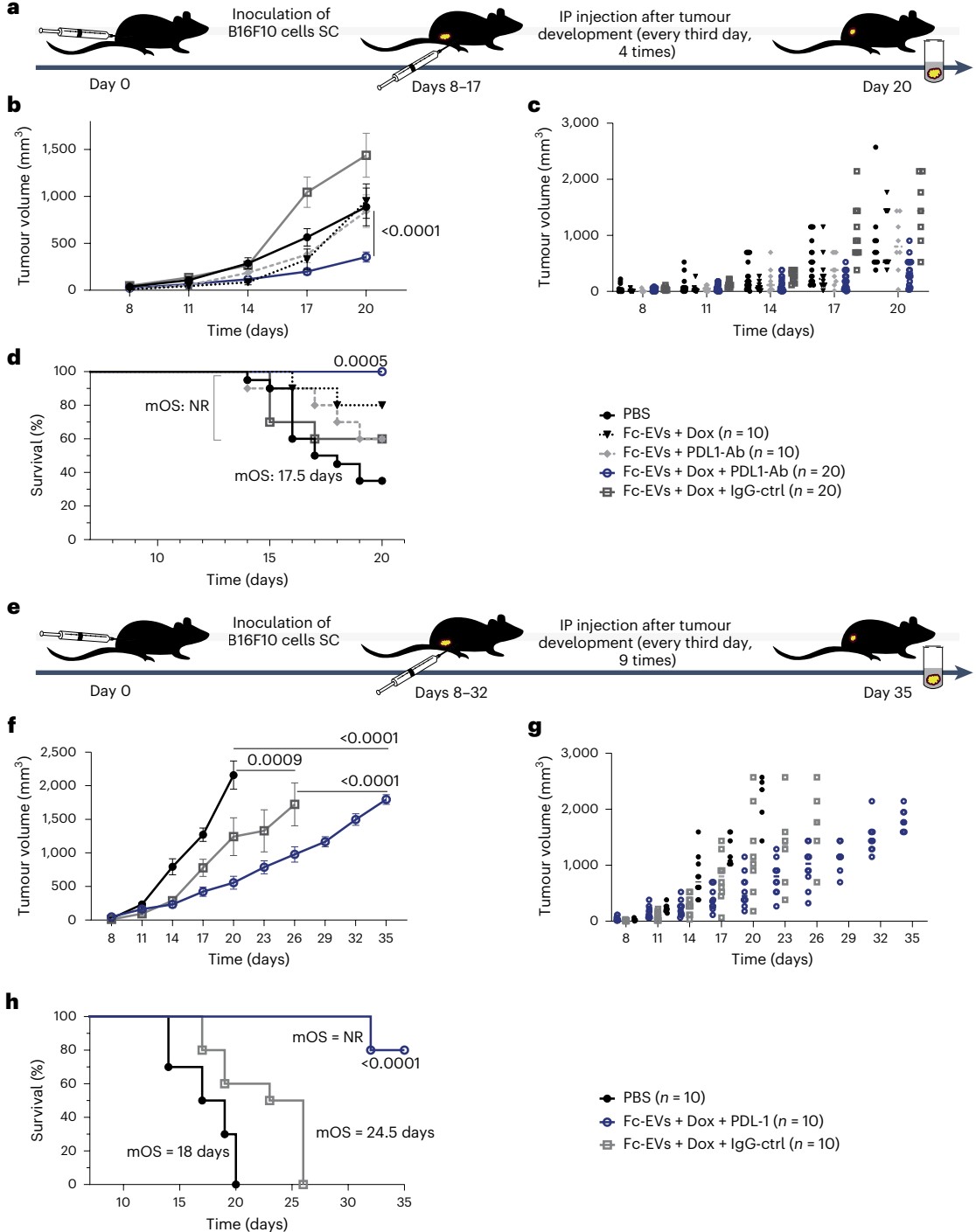

**Fig. 7 | Tumour treatment with Dox-loaded Fc-EVs displaying PD-L1-Ab. a**, The experimental set-up. Mice were inoculated with malignant melanoma (B16F10) cells on day 0. After tumour formation, mice were treated every third day for four cycles until day 20. d, day(s). **b–d**, Tumour volume (**b**, average volume and **c**, individual tumour volume of live mice) and survival (**d**) over time for mice treated with combinations consisting of Dox-loaded Fc-EVs (Fc-EVs + Dox) without Ab or with PD-L1-Ab (atezolizumab, Fc-EVs + Dox + PD-L1-Ab) or with control-Ab (Fc-EVs + Dox + IgG-ctrl) compared to Fc-EVs with PD-L1-Ab without Dox (Fc-EVs + PD-L1-Ab) or mock treatment (PBS), displaying significant improvement only for Fc-EVs + Dox + PD-L1-Ab. Mean overall survival (mOS) was not reached (NR) for any group except PBS. **e**, The set-up for prolonged experiment. After tumour formation, mice were treated every third day for nine cycles until day 35. **f,g**, Tumour volume (**f**, average volume and **g**, individual tumour volume of live mice) over time for live mice treated with Dox-loaded Fc-EVs with PD-L1-Ab (Fc-EVs + Dox + PD-L1-Ab) or control-Ab (Fc-EVs + Dox + IgG-ctrl) compared to mock treatment (PBS), displaying significant improvement for Fc-EVs + Dox + PD-L1-Ab. **h**, Survival over time for the same treatment as in **f** and **g**, showing mOS for the different treatment groups, which was not reached on day 35 for Fc-EVs + Dox + PD-L1-Ab. All data are shown as mean ± s.d. n = 10–20 mice (as indicated per group). Statistical significance was calculated using two-way ANOVA with Dunnett's (**b** and **c**) or Tukey's (**f**) post-test or Mantel–Cox test (**d** and **h**) compared with each value; P values are indicated above the plots throughout.

studies on the Fc-EV concept will hopefully explore the therapeutic potential further and possibly translate it to clinical use.

## Methods

### Cell culture

FreeStyle 293F cells were cultured in FreeStyle medium (FreeStyle 293; Gibco Life Technology) supplemented with 1% Antibiotic Antimycotic (Gibco Life Technology) and 4.5 g l$^{-1}$ pyruvate (Invitrogen). HEK293T, SKBR-3 and B16F10 cells were cultured in DMEM high glucose medium (DMEM; Gibco Life Technology), supplemented with 1% Antibiotic Antimycotic (Invitrogen) and 10% fetal bovine serum (FBS) (Gibco). Mycoplasma testing was routinely performed, and all cell lines were maintained at 37 °C with 5% CO$_2$ in a humidified atmosphere.

### Production of engineered EVs

**Plasmid constructs and cloning.** For the different Fc-binding constructs, the amino acid sequence for all nine EV-sorting domains (Supplementary Fig. 12), nine Fc binders (listed in Supplementary Fig. 13), mNG (Supplementary Fig. 13) and nLuc (Supplementary Fig. 13) were confirmed by Uniprot. They were subsequently synthesized (Integrated DNA Technologies) as gene blocks and cloned into a lentiviral p2CL9IPwo5 backbone downstream of the SFFV promoter using NotI and EcoRI and upstream of an internal ribosomal entry site-puromycin resistance complementary DNA cassette. To create the plasmid-expressing mNG fused to the C- or N-terminus of the EV-sorting domains, mNG was subcloned into the p2CL9IPw5-EV (plasmid was kindly given by H. Hanenberg, University Hospital Essen, Germany) sorting domain construct using NotI and SphI or AgeI and EcoRI, respectively. The nLuc was subcloned into the p2CL9IPw5-EV-sorting domain construct using NotI and SphI. The Fc binders were then cloned into the p2CL9IPw5-EV-sorting domains using Kpn2I and Bsp119I. Before using the constructs, all plasmid constructs were confirmed by Sanger sequencing (Eurofins Genomics).

### Virus production and transduction

Lentiviral supernatants were produced as described previously[16]. In brief, HEK-293T cells were co-transfected with the p2CL9IPw5 construct, including CD63-Fc fused either to mNG or nLuc proteins, the human foamy virus envelope construct pcoPE and the helper construct pCD/NL-BH using the transfection reagent JetPEI (Polyplus). Eighteen hours post transfection, gene expression from the human cytomegalovirus immediate-early gene enhancer/promoter was induced with 10 mM sodium butyrate (Sigma-Aldrich) for 6 to 7 h before fresh media was added to the cells, and the supernatant was collected after 22 h. Viral particles were then pelleted at 25,000 g for 90 min at 4 °C. The pellet was resuspended in 1 ml of Iscove's modified Dulbecco's media supplemented with 20% FBS and 1% antibiotic antimycotic after the supernatant had been removed. Until use, the viruses were kept at 80 °C. The FreeStyle 293F cells were then transduced overnight with the virus and passaged a minimum of five times under puromycin selection to make stable cell lines.

### EV isolation

EVs were isolated, as described in a previous publication[13]. In brief, once a concentration of $2 × 10^6 ± 2 × 10^5$ FreeStyle 293F cells per ml was obtained, the condition media was collected and centrifuged at 700 RCF for 5 min to remove cells. The pellet was discarded, and the condition media was centrifuged for an additional 10 min at 2,000 g to remove cell debris and additional larger contaminants. Subsequent purification by 0.22 μm filtration was performed (Nalgene Rapid-Flow; Thermo Fisher Scientific), and the EVs were concentrated by tangential flow filtration (KrosFlo Research 2i TFF system; Spectrum labs) at a flow rate of 100 ml min$^{-1}$ and a transmembrane pressure around 20,684 kPa using a 300 kDa polyether sulfone hollow fibre filter (MidiKros, 370 cm$^2$ surface area, SpectrumLabs). The sample

was diafiltered with twice the original condition media volume using 0.22 μm filtered (Nalgene Rapid-Flow; Thermo Fisher Scientific) 0.1 M PBS and collected at a final volume of 20–35 ml. The concentrate was purified by 0.22 μm suction filtration (Nalgene Rapid-Flow; Thermo Fisher Scientific), and the filtrate was further concentrated by ultra-filtration at 4,000 RCF in 10 kDa spin columns (Amicon Ultra-15; Millipore) to a final volume of 400–600 μl. The EVs were stored in PBS-HAT (HEPES, Albumin, Trehalose) buffer at −80 °C until further use, as described in ref. 47.

### Nanoparticle tracking analysis

NTA was performed with the NS500 nanoparticle analyser (NanoSight) to measure the size distribution and concentration of EVs as described in ref. 6. The samples were diluted in 0.1 M PBS to achieve a particle count of between $2 × 10^8$ and $2 × 10^9$ per ml. The camera focus was adjusted to make the particles appear as sharp dots. Using the script control function, five 30 s videos for each sample were recorded, incorporating a sample advance and a 5 s delay between each recording. The analysis was performed with the screen gate set at 10 and the detection threshold at 7, with all remaining settings set at automatic.

### Transmission electron microscopy

Purified Fc-EVs or ctrl-EVs were incubated with 1 μl of 1% BSA diluted in PBS for 5 min. Rabbit anti-goat 10 nm antibody conjugated with gold nanoparticles (BBI Solutions) was added and incubated for 45 min. Finally, 5 μl of labelled EVs were added onto glow-discharged formvar-carbon-type-B-coated electron microscopy grids (Ted Pella) for 3 min. The grid was dried with filter paper, washed twice with distilled water and blotted dry with filter paper. After the wash, the grid was stained with 2% uranyl acetate in double-distilled H$_2$O (Sigma-Aldrich) for 10 s and filter paper dried. The grid was air dried and visualized on a transmission electron microscope (Tencai 10).

### Protein detection by western blot

The presence of the EV-associated tetraspanin CD63 fused to the Nano-luc (Nluc) protein as well as the intravesical proteins Alix and TSG101 was examined by western blot. About $2 × 10^6$ HEK293 cells from the production culture were pelleted at 300 RCF for 5 min, washed with cold PBS and pelleted once more. The cell pellet and $1 × 10^{10}$ EVs were lysed separately using 100 μl of radioimmunoprecipitation buffer (BioRad). Both samples were incubated on ice for 30 min and vortexed for 10 s each fifth minute. Lipids and additional large contaminants were removed from the cell lysis by centrifugation at 12,000 RCF for 12 min at 4 °C. About 24 μl of the supernatant was transferred to a new tube on ice. The collected supernatant and 24 μl EVs were mixed with 8 μl loading buffer (10% glycerol, 8% sodium dodecyl sulfate, 0.5 M dithiothreitol and 0.4 M sodium carbonate). The protein was further denatured to its primary structure by incubation at 65 °C for 5 min before being loaded onto the NuPAGE+ (Invitrogen, Novex 412% Bis–Tris gel) and run at 120 V for 2 h. The proteins were transferred by iBlot system (iBlot 2 Dry Blotting System; Invitrogen) for 7 min to an iBlot membrane (iBlot 2 Transfer Stacks; Invitrogen). The membrane was treated by blocking buffer (Odyssey Blocking Buffer; LI-COR Biosciences) at room temperature for 60 min to avoid unspecific binding and later incubated overnight at 4 °C with newly prepared primary antibodies (anti-CD63 (ab134045, Abcam) diluted 1:1,000, anti-Alix (MA1-83977, Thermofisher) diluted 1:1,000, anti-nLuc (N7000, Promega) diluted 1:1,000 and anti-Tsg101 (ab30871, Abcam) diluted 1:1,000. The membranes were washed 4 times with tris-buffered saline with 0.1% tween for 5 min each on a shaker and then incubated with secondary antibody (goat anti-mouse (C00322) diluted 1:10,000, goat anti-rabbit (C90827-25) diluted 1:10,000) at room temperature for 1 h. The washing was repeated with one additional PBS wash, and the results were visualized by both 700 and 800 nm channels of an infrared imaging system (LI-COR Odyssey CLx).

## Stability of antibody display of Fc-EVs

To test the stability of antibody-functionalized EVs, Fc-EVs (with intravesicular mNG reporter) were first decorated with human IgG1-APC antibody (Miltenyi, catalogue number 130-11-434). Specifically, $1 \times 10^{11}$ Fc-EVs were mixed with 25 µg antibody in a total volume of 0.2 ml, followed by 2 h incubation at 37 °C. After dilution with 0.2 ml Dulbecco's phosphate-buffered saline (DPBS), EVs were purified using qEV size exclusion chromatography columns (Izon; SP1) according to manufacturer's instructions. To examine the stability of engineered EVs in plasma, $1 \times 10^8$ APC-labelled EVs (in 20 µl) were incubated with 10 µl plasma (from healthy human or mice) in 96-well V-bottom plates at 37 °C. After 1, 10 or 30 min, the samples were diluted with DPBS by 1,500-fold and analysed using CellStream flow cytometer as described below. Stability in the presence of other antibodies was similarly tested by incubating $1 \times 10^8$ APC-labelled EVs with 25, 50 or 250 ng phycoerythrin (PE)-conjugated antibody for 30 min. The percentage of APC$^+$mNG$^+$ (double positive) among total mNG$^+$ EVs was calculated. The following PE-conjugated antibodies were included: hIgG1-PE (Miltenyi, catalogue number 130-113-438), hIgG4-PE (BioLegend, catalogue number 403704), mIgG1-PE (Miltenyi, catalogue number 130-113-200) and mIgG2-PE (Miltenyi, catalogue number 130-092-215).

## Binding affinity of Fc-EVs with antibody

To characterize the affinity of Fc-EVs with antibodies, $1 \times 10^8$ EVs (with intravesicular mNG reporter, in 20 µl) were incubated with different amount of PE-conjugated antibody (0.0125–10 µg ml$^{-1}$; in 10 µl) in 96-well V-bottom plates at 37 °C. After 2 h, the samples were diluted with DPBS by 1,500-fold and analysed using CellStream as described below. The percentage of APC$^+$mNG$^+$ (double positive) among total mNG$^+$ EVs was calculated and fitted against antibody concentration using the one-site total binding model from GraphPad. The predicted equilibrium dissociation constant $K_d$ was presented for each type of antibody if available. The following PE-conjugated antibodies were included: hIgG1-PE (Miltenyi, catalogue number 130-113-438), hIgG4-PE (BioLegend, catalogue number 403704), mIgG1-PE (Miltenyi, catalogue number 130-113-200) and mIgG2-PE (Miltenyi, catalogue number 130-092-215).

## Antibody quantification per EVs

To determine the number of antibodies that bind to Fc-EVs, the Fc-EVs (with intravesicular mNG reporter) were first decorated with human IgG1-APC antibody (Miltenyi, catalogue number 130-11-434) or mouse IgG2b (Miltenyi Biotec 130-092-215) and ctrl-EVs (with intravesicular mNG reporter) decorated with human IgG1-APC antibody (Miltenyi, catalogue number 130-11-434) or mouse IgG2b (Miltenyi Biotec 130-092-215). Specifically, $1 \times 10^{11}$ Fc-EVs or ctrl-EVs were mixed with 25 µg antibody in a total volume of 200 µl, followed by 2 h incubation at 37 °C. After dilution with 200 µl DPBS, Fc-EVs were purified using qEV columns (Izon; SP1) according to manufacturer's instructions. For quantification of antibodies per EVs, the SPP was performed[24]. Briefly, fluorescence intensity (Supplementary Fig. 14a) fluctuations were recorded by observing free diffusion of fluorescently labelled EVs through the observation volume. The histogram of fluorescent brightness of antibodies per single EV was converted into the histogram in terms of antibody number per particle (Supplementary Fig. 14b) using the fluorescence brightness of free antibody, which was quantified by fluorescence correlation spectroscopy (Supplementary Fig. 14c–e). Curves were fitted with the following three-dimensional diffusion, and molecular brightness was quantified:

$$G(\tau) = \frac{1}{\langle N \rangle}\left(1 + \frac{\tau}{\tau_D}\right)^{-1}\left(1 + \frac{\tau}{AR^2 \times \tau_D}\right)^{1/2}$$

where $G(\tau)$ is a correlation function, $\tau$ is delay time, $\tau D$ is diffusion time, AR is for aspect ration of the observation volume and $N$ is an average number of molecules within the observation volume. Molecular brightness is quantified as counts per second per particle. SPP was performed using the set-up for fluorescence correlation spectroscopy on a Zeiss LSM 780 microscope. A 488 nm argon ion laser was used for mNG and a 633 nm He–Ne laser was used for APC detection. A 40 × 1.2 NA water immersion objective was used to focus the light. The laser power was set to 0.1–0.5% of the total laser power, corresponding to 2–10 µW. The emission detection windows were set to 490–560 for mNG and 650–700 for APC.

## Flow cytometry

Validation of purified EVs was performed by flow cytometry using two different techniques described in the following sections.

**Multiplex bead-based flow cytometry analysis.** Multiplex bead-based flow cytometry analysis (MACSPlex Exosome Kit, human, Miltenyi Biotec) was implemented to characterize the general surface protein composition of Fc-EVs and to assess the binding of specific antibodies to the Fc receptor expressed on the EVs. Assays were performed on the basis of a previously described protocol[16]. In brief, EVs were used at an input dose of $1 \times 10^9$ NTA-based particles per assay, diluted with MACSPlex buffer to a total volume of 120 µl and incubated with 15 µl of MACSPlex exosome capture beads overnight in wells of a pre-wet and drained MACSPlex 96-well 0.22 µm filter plate on an orbital shaker at 450 rpm at room temperature. The beads were washed with 200 µl MACSPlex buffer, and the liquid was removed by applying a vacuum (Sigma-Aldrich, Supelco PlatePrep; −100 mBar). For counterstaining of captured EVs, either a mixture of APC-conjugated anti-CD9, anti-CD63 and anti-CD81 detection antibodies (supplied in the MACSPlex kit, 5 µl each) or AlexaFluor647-conjugated human IgG Fc fragments (The Jackson Laboratory, catalogue number 009-600-008, 100 ng) was added to each well in a total volume of 135 µl, and the plates were incubated on an orbital shaker at 450 rpm for 1 h at room temperature. Next, the samples were washed twice, resuspended in MACSPlex buffer and analysed by flow cytometry with a MACSQuant Analyzer 10 flow cytometer (Miltenyi Biotec). FlowJo (v.10.6.2, FlowJo) was used to analyse flow cytometry data.

**Single-vesicle imaging flow cytometry.** Single-vesicle imaging flow cytometry was performed as described previously[11,47] by using either an Amnis ImageStream X Mk II or an Amnis Cellstream instrument (Amnis/Luminex). In brief, $2.5 \times 10^8$ EVs (NTA-based particles) were stained (unless indicated otherwise) in a total volume of 25 µl with 8 nM of fluorescent antibodies against the tetraspanins CD9, CD63 and CD81 or recombinant antibodies (isotypes control) (REA(S))-APC human IgG isotype control antibodies (Miltenyi Biotec). EVs were incubated overnight at room temperature in the dark with a subsequent dilution to a concentration of $1 \times 10^7$ EVs per ml in a final volume of 100 µl before data acquisition. Unstained samples and non-EV-containing samples incubated with antibodies were included as controls, respectively. PBS-HAT buffer was used as diluent for all steps[47]. The results were analysed with FlowJo software (v.10.7.2; FlowJo LLC).

**Analysis of tissue, blood and tumour single-cell suspensions by flow cytometry.** To remove cell aggregates, appropriate dilutions of tissue and tumour cell suspensions (1:10 in PBS/1% FBS for tumour, liver and spleen cell suspensions; no dilution for peripheral blood mononuclear cell (PBMC) suspensions and lymph node cell suspensions) were prepared and filtered through a 40 µm cell strainer cap directly into a 5 ml round bottom tube (Corning) for flow cytometry to remove aggregates. All samples were subjected to four different antibody staining protocols to detect PD-L1 simultaneously with T cells (CD3, CD8), B cells (CD45R/B220), monocytes (CD11b) or macrophages (F4/80), respectively. For all stainings, 25 µl of cell suspension was added to each well of a 96-well V-bottom microplate (Sarstedt) and 25 µl

of PBS/1% FBS was added containing a mix of the respective antibody combinations. Cells were stained for 30 min at 4 °C in the dark. To wash off unbound antibodies, wells were filled up with PBS/1% FBS, and the plate was centrifuged at 900 $g$ for 5 min. Cell pellets were resuspended in 90 μl PBS/1% FBS and kept in the dark at 4 °C until flow cytometric measurement on a MACSQuant Analyzer 10 instrument equipped with three lasers (405 nm, 488 nm, 635 nm, Miltenyi Biotec) using the 96-well plate experiment layout. For live/dead cell discrimination 10 μl of a 10× DAPI (4′,6-diamidino-2-phenylindole, Sigma-Aldrich) stock solution was added to each well. The data were analysed using the FlowJo Analysis Software (v. 10.8, FlowJo LLC, BD Biosciences). Antibodies (fluorophore, final dilution, clone and manufacturer) were rb-α-ms PD-L1 (APC, 1:100, 485, Sino Biological), rt-α-ms CD3 (PerCP/Cy5.5, 1:100, 17A2, BioLegend), rt-α-ms CD8a (BV510, 1:100, 53-6.7, BioLegend), rt-α-ms CD45R/B220 (BV510, 1:100, RA3.6B2, BioLegend), rt-α-ms CD11b (BV510, 1:100, M1/70, BD Biosciences) and rt-α-ms F4/80 (BV510, 1:100, T45-2342, BD Biosciences).

### Size exclusion chromatography

To confirm the binding affinity of Fc-EVs with antibodies, $1 \times 10^{11}$ mNG-Fc-EV or control mNG-EV was incubated for 2 h at 37 °C in 400 μl of 0.22 μm filtered PBS with 2.5 μg REA(S)-APC human IgG isotype control antibodies (Miltenyi Biotec). Following this, the samples were fractionated using 70 nm, 500 μl qEV columns (qEV original; IZON Science LTD), yielding 25 fractions of 1 ml per fraction. The fractions were analysed by fluorometer (SpectraMax i3x; Molecular Devices LLC).

### Cell-uptake experiment

For Huh-7 cells uptake of human IgG Fc fragment bound to Fc-EVs, 100,000 Huh-7 cells were seeded overnight in a 24-well plate before Fc-EV treatment. About $1 \times 10^{11}$ mNG$^+$ ctrl-EV or Fc-EVs were incubated with 2 μg of human IgG Fc fragment (AF647 conjugated) overnight at 4 °C with rotation. The next day, Huh-7 cells were treated with the Fc fragment, with or without ctrl-EV or Fc-EVs and incubated at 37 °C with 5% CO$_2$ for 2 h. The Huh-7 cells were then analysed in a motorized Olympus IX-81 inverted-fluorescence microscope equipped with an XM10-monochrome camera and narrow band-filter cube for UV (DAPI), green (GFP) and red (APC) excitation.

For B16F10, stimulated with or without 40 ng ml$^{-1}$ of IFNγ (PMC4031, Gibco Lifetechnology) and SKBR-3 cell lines, the cells were seeded in a 96-well plate format at $1 \times 10^4$ and $2.5 \times 10^4$ cells per well, respectively. The cells were incubated and maintained at 37 °C with 5% CO$_2$ in a humidified atmosphere for 24 h before treatment. After that, the cells were treated for 2 h with either mNG-Fc-EVs + anti-PD-L1 (atezolizumab) or anti-HER2 (trastuzumab) antibodies or control mNG-EVs + anti-PD-L1 (atezolizumab) or anti-HER2 (trastuzumab) antibodies. The cells were then washed with PBS, trypsinized and resuspended in fresh DMEM medium for flow cytometry using a MACSQuant Analyzer 10 flow cytometer (Miltenyi Biotec). FlowJo (v. 10.6.2, FlowJo) was used to analyse flow cytometry data.

To assess the effect on EV uptake by pretreatment of HER2-Ab, $25 \times 10^3$ SKBR-3 cells were seeded in a 96-well plate format per well for 24 h in DMEM high-glucose medium (DMEM; Gibco Life Technology), supplemented with 1% antibiotic antimycotic (Invitrogen) and 10% FBS (Gibco). The plate was maintained at 37 °C with 5% CO$_2$ in a humidified atmosphere. The cells were then treated with 2.5 μg of anti-HER2 (trastuzumab) or left untreated for 2 h. mNG-Fc-EVs + 2.5 μg of anti-HER2 (trastuzumab) or Fc-EVs alone or mNG-Fc-EVs + 2.5 μg of human IgG isotype control antibody (catalogue number 12000C, Thermo Fisher) were subsequently added to the cells for another 2 h. The cells were subsequently washed with PBS, trypsinized and resuspended in fresh DMEM medium for flow cytometry using a MACSQuant Analyzer 10 flow cytometer (Miltenyi Biotec). FlowJo (v. 10.6.2, FlowJo) was used to analyse flow cytometry data.

### Kidney and liver function tests

Tests for creatinine (Abcam: ab65340) and urea (Abcam: a234052) for kidney function were performed based on the manufacturer's instructions. Briefly, serum samples were separated by 10 kDa and spin-filtered at 10,000 $g$ for 10 min at 4 °C. After collecting the filtrate, 2–10 μl of samples was added to a 96-well plate; the final volume was adjusted to 50 μl with distilled H$_2$O, and absorbance was measured by fluorometer (SpectraMax i3x; Molecular Devices LLC) at 570 nm and 505 nm, respectively. Aspartate aminotransferase (Abcam: ab105135) and alanine transaminase (Abcam: ab105134) for liver function tests were performed based on the manufacturer's instructions. In brief, serum samples were directly diluted in the assay buffer. Samples were prepared at up to 50 μl per well with assay buffer in a 96-well plate. Absorbance was measured by fluorometer (SpectraMax i3x; Molecular Devices LLC) at 450 nm and 570 nm, respectively.

### Fluorescence microscopy

B16F10 cells were seeded at 20,000 cells per well in 8-well glass-bottom Nunc Lab-Tek II Chamber slide (Thermo Scientific). Cells were allowed to adhere for 24 h and then stimulated with or without 40 ng ml$^{-1}$ of IFNγ (PMC4031, Gibco Lifetechnology). Cells were treated for 2 h with labelled antibody (anti-PD-L1-APC, Sinobiological, catalogue number 50010-R485-A) or mNG$^+$ Fc-EVs alone, or mNG$^+$ Fc-EVs with PD-L1-Ab (atezolizumab or anti-PD-L1-APC) as described above. After the treatment period, cells were washed once with PBS and then fixed with 4% paraformaldehyde (Fisher Scientific) for 10 min at room temperature. Both PBS and fixation solutions were warmed up to 37 °C, and the pipetting was conducted swiftly to avoid sample drying. Following fixation, nuclei were stained with DAPI (Fisher Scientific) for 15 min. Stained cells were then washed three times with PBS. Images were acquired using a confocal microscope (A1R confocal, Nikon) and analysed by the NIS-Elements software (v. 5.21.00, Nikon).

### Nanoimager microscopy

For the antibody to Fc-EVs conjugation, 5 μg of antibody was mixed with $1 \times 10^{11}$ Fc-EVs in PBS 0.2% human albumin serum in a final volume of 100 μl, overnight on a thermoblock plate at 25 °C with a 300 r.p.m. rotation and protected from light. For sample purification, a qEV column was initially equilibrated with 2 ml volume of PBS. Following reaction completion, the samples were purified by adding the full 100 μl of the reaction mixture onto a qEV column (Izon qEVsingle Single-use Columns, Fisher Scientific, catalogue number 15928090). Fraction collection was performed by adding 3 ml of PBS to the column and eluting by counting eight drops per fraction, on a total of 21 fractions. For imaging, each well of a μ-Slide 18 Well Glass Bottom slide (IbiDi, catalogue number 81817) was washed with distilled water twice followed by a wash with potassium hydroxide 1 M and a final wash with distilled water for surface cleaning. For surface coating, 100 μl of poly-L-lysine solution (Sigma, catalogue number 25988-63-0) was added and incubated at 37 °C for 2 h. Upon incubation period completion, poly-L-lysine solution was carefully removed with a pipette, and the sample of antibody-Fc-EVs was added (50 μl); the plate was left overnight at 4 °C. Before imaging, the plate was allowed to reach room temperature. The high-resolution experiment was performed using a Nanoimager S Mark II microscope from ONI (Oxford Nanoimaging) equipped with a ×100, 1.4 NA oil immersion objective, an XYZ closed-loop piezo 736 stage and triple emission channels split at 488, 555 and 640 nm. For the evaluation of antibody-Fc-EV conjugation the manufacturer's protocol was used.

### Loading Fc-EVs with Dox

Electroporation was conducted based on our recently published protocol[39]. Dox (doxorubicin hydrochloride, Sigma-Aldrich; Merck) was diluted to a concentration of 10 mM with ultrapure water, to avoid aggregation at later stages. The Dox was mixed with EVs diluted with

0.22 µm filtered (Nalgene Rapid-Flow; Thermo Fisher Scientific) 0.1 M PBS and incubated at 4 °C for 30 min. Prepared electroporation buffer (400 mM sucrose in PBS) was added to the EVs in a 1:1 ratio, and 400 µl of the sample was electroporated in 0.4 cm cuvettes by exponential pulse using an electroporation system (GenePulser Xcell; BioRad). The sample was then incubated at 37 °C for 30 min, and the EVs were subsequently isolated by SEC, using 70 nm 500 µl qEV columns (qEVoriginal; IZON Science LTD), collecting the 4th and 5th millilitres of elute. The elute was then concentrated to a final volume of 200–300 µl, using 10 kDa spin filters (Amicon Ultra; Millipore) spun at 4,000 RCF. Dox was analysed by fluorometer (SpectraMax i3x; Molecular Devices LLC), and the concentrations were determined in relation to a standard curve starting at 100 µM halving 12 times to 40 mM and blanks of ultrapure water and PBS, using the dedicated software (SoftMax Pro v.7; Molecular Devices LLC). The samples were portioned at 90 µl, excited at 488 nm and read at 530 nm. Effective loading was calculated as a function of the concentration of Dox (mM per billion EVs).

### Stability of encapsulated Dox in plasma

Dox was loaded into Fc-EVs by electroporation, as described above. The electroporated Fc-EVs were purified by SEC to remove unloaded Dox. After SEC, the sample was concentrated by a 2 ml 10 kDa spin filter. The concentrated samples were mixed with 1 ml of mouse plasma and incubated at three time points (1, 15 and 30 min) at 37 °C. The samples were subsequently purified by SEC and concentrated using a 2 ml 10 Kd spin filter. Encapsulated Dox was analysed by fluorometer (SpectraMax i3x; Molecular Devices LLC), and the concentrations were determined in relation to a standard curve starting at 100 µM halving 12 times to 40 mM and blanks of ultrapure water and PBS, using the dedicated software (SoftMax Pro v.7; Molecular Devices LLC). The samples were portioned at 90 µl, excited at 488 nm and read at 530 nm.

### Viability assay

The viability of the cells after treatment was assessed by 80 µl per well Cell Titre Glo (CellTiterGlo; Promega Biotech) following the provided protocol. About $1 \times 10^4$ B16F10 cells were seeded in a 96-well plate 24 h before treatment, and the plate was read 48 h post treatment, with a luminometer (GloMax 96 Microplate Luminometer; Promega Biotech) using the predesigned protocol provided by the manufacturer. The plate was read each third minute until the signal stabilized.

### Animal experiments

All of the animal experiments were performed in accordance with ethical permissions approved by the Swedish Local Board for Laboratory Animals and designed to minimize the suffering and pain of the animals. The animals were supplied by Scanbur or Janvier Labs.

**Malignant melanoma model.** The tumours were established as previously described[48]. Briefly, female C57BL/6 (20 ± 2 g) mice were subcutaneously implanted with $5–7 \times 10^5$ B16F10 cells on day 0. They were monitored daily and developed palpable tumours within 7 days. The mice were killed at pre-decided time points, if the tumour size exceeded 1,500 mm³ or if the mice exceeded the scoring of pre-set humane end points. For distribution experiments, the mice ($n = 7–10$) were injected intravenously (if not otherwise indicated) when the tumours had reached 1,000 mm³ (on day 14 ± 2) with $1 \times 10^{11}$ Fc-EVs alone or following 4 °C incubation over-night with 2.5 µg anti-PD-L1 (atezolizumab) or 2.5 µg REA(S)-APC human IgG isotype control antibody (Miltenyi Biotec). After the injection the mice were killed for tissue and blood collection at different time points (15 min, 30 min, 1 h, 3 h, 24 h, 48 h or 72 h). For luminescent reading, nLuc positive Fc-EVs were used, whereas mNG⁺ Fc-EV were used for flow cytometry applications (30 min time point only). The tissues were processed as described in ref. 29, with lysed tissue and nLuc⁺ EV input being analysed for nanoluc luminescence or for single-cell suspension for flow cytometry, as described

below. For tumour treatment experiments with repetitive injections, mice were treated by intraperitoneal injections every third day starting from 8 days after inoculation of mice with palpable tumours. In the short-term experiment all mice were killed no later than day 20. The treatment groups, $n = 10–20$, were as follows:

1. PBS
2. $1 \times 10^{11}$ Fc-EVs alone
3. 25 µg anti-PD-L1 (atezolizumab)
4. 25 µg IgG human isotype control (catalogue number 12000C, Thermo Fisher),
5. 50 µg Dox (Sigma-Aldrich)
6. Fc-EVs + PD-L1-Ab ($11 \times 10^{11}$ Fc-EVs which had been incubated for 2 h at 37 °C in 0.22 µm filtered PBS with 25 µg anti-PD-L1 (atezolizumab))
7. Fc-EVs + Dox ($1 \times 10^{11}$ Dox-loaded Fc-EVs alone (prepared as described above)
8. Fc-EVs + Dox + PD-L1-Ab ($11 \times 10^{11}$ Dox-loaded Fc-EVs which had been incubated for 2 h at 37 °C in 0.22 µm filtered PBS with 25 µg anti-PD-L1 (atezolizumab))
9. Fc-EVs + Dox + PD-L1-Ab ($11 \times 10^{11}$ Dox-loaded Fc-EVs which had been incubated for 2 h at 37 °C in 0.22 µm filtered PBS with 25 µg anti-PD-L1 (atezolizumab))

In the long-term experiment, the mice were killed no later than day 35 and were given treatment, $n = 10$, using groups 1, 8 and 9 as described above. The mice were observed daily, and tumour size was measured every third day using a calliper. Blood was collected when possible before killing the animals.

**Preparation of single-cell suspensions from tissue.** Mice were killed, and organs (liver, spleen, lymph nodes) were surgically resected and kept on ice submerged in appropriate amounts of PBS supplemented with 1% FBS until further processing. Next, tissues were homogenized by mashing the cells through a 70 µm Falcon cell strainer (Thermo Fisher Scientific) with a syringe plunger. The cell strainer was washed several times with ice-cold PBS/1% FBS buffer to elute all remaining cells into a 50 ml conical tube. Cell suspensions were filled up to 25 ml with ice-cold PBS/1% FBS. Here, 10% of liver suspensions were transferred into a new tube with sufficient amounts of PBS/1% FBS for further processing. Next, suspensions were centrifuged at 500 *g* for 5 min to collect the cells, and the supernatants were carefully decanted. Liver cells and spleen cells were resuspended in 500 µl 1× BD Pharm Lyse lysing solution (BD Biosciences) and incubated for 10 min at room temperature in the dark to lyse remaining red blood cells (RBCs). To stop the lysis, suspensions were filled up to 30 ml with ice-cold PBS/1% FBS and centrifuged at 500 *g* for 5 min. The supernatants were removed carefully by pipetting. Pellets from lymph node cell suspensions were not subjected to RBC lysis but processed in parallel in all subsequent steps. Next, cell pellets were washed by resuspension in 10 ml PBS/1% FBS, transferred to fresh 15 ml conical tubes and centrifuged at 500 *g* for 5 min. Supernatants were carefully removed, and pellets were resuspended in 500 µl PBS/1% FBS. The final cell suspensions were kept at 4 °C until further processing for flow cytometry.

**Preparation of PBMCs from blood.** To obtain suspensions of erythrocyte-depleted PBMCs, mice were killed, and 200 µl blood was collected into BD Microtainer tubes (BD Biosciences) containing EDTA by heart puncture. About 180 µl EDTA–blood was transferred to a 1.5 ml tube and centrifuged at 500 *g* for 7 min at 4 °C. The supernatant was removed, cells were gently resuspended in 200 µl FBS/1% PBS, transferred to 15 ml conical tubes containing 500 µl 1× BD Pharm Lyse lysing solution (BD Biosciences) and incubated for 10 min at room temperature in the dark to lyse RBCs. To stop the lysis, suspensions were filled up to 10 ml with ice-cold PBS/1% FBS and centrifuged at 500 *g* for

5 min. After supernatants were removed carefully, the RBC lysis step was repeated for 8 min, followed by lysis stop and centrifugation as before. The final cell pellet was resuspended in 500 µl PBS/1% FBS and kept at 4 °C until further processing for flow cytometry.

**Preparation of single-cell suspension from B16F10 cell-engrafted tumours.** After mice were killed, tumours were surgically resected, placed into a 60 mm dish, and submerged in room temperature PBS. The tissue was minced with a scalpel to obtain small pieces (1–3 mm³), and pieces were transferred into a 50 ml conical tube containing 5 ml pre-warmed B16F10 cell culture medium. The tissue was centrifuged at 100 $g$ for 5 min at room temperature, and the supernatant was removed carefully. Next, tissue pieces were submerged in 4.7 ml pre-warmed B16F10 cell culture medium and 250 µl 20× collagenase II (2,500 collagen digestion unit ml$^{-1}$ (20 mg ml$^{-1}$, Sigma-Aldrich) and 50 µl 100× DNase I (10,000 Kunitz per ml, Sigma-Aldrich) were added to final concentrations of collagenase II and DNase I at 1 mg ml$^{-1}$ and 100 Kunitz per ml, respectively. The tubes were fixed on a rocking platform and incubated at 37 °C for 60 min. After digestion, the cell suspensions were carefully pipetted for at least 25 times with a 10 ml serological pipette until the suspensions appeared homogenous without apparent tissue pieces. Next, the suspensions were filtered through a 70 µm Falcon cell strainer (Thermo Fisher Scientific) into a fresh 50 ml conical tube, followed by a subsequent filtration through a 40 µm Falcon cell strainer (Thermo Fisher Scientific) into another fresh 50 ml conical tube. The cell strainer was washed slowly with 10 ml of pre-warmed B16F10 cell culture medium to collect remaining cells. Cell suspensions were centrifuged at 500 $g$ for 10 min to pellet the cells, and the supernatant was removed carefully. The cell pellets were resuspended in 500 µl cold PBS/1% FBS and kept at 4 °C until further processing for flow cytometry.

**Human breast cancer HER2 positive model.** Biodistribution studies were performed in SKBR-3-bearing Swiss nude mice as previously described[35]. Briefly, 35 six-week-old female Swiss nude mice (Charles River) were orthotopically grafted (mammary fat pad) with $5 \times 10^6$ SKBR-3 cells in 60% matrigel (CLS354234-1EA, Sigma-Aldrich). They were monitored daily and developed palpable tumours within 2 months. The mice were killed at pre-decided time points if the tumour size exceeded 1,500 mm³ or if the mice exceeded the scoring of pre-set humane end points. The mice ($n = 5–9$) were injected intravenously on day 60 with $1 \times 10^{11}$ Fc-EVs, which had been incubated for 2 h at 37 °C in 0.22 µm filtered PBS with 25 µg anti-HER2 (trastuzumab) or 25 µg human IgG isotype control antibody (catalogue number 12000C, Thermo Fisher). Thirty minutes after the injection, the mice were killed for tissue and blood collection. The tissues were processed as described in ref. 34, with lysed tissue and nLuc$^+$ EV input being analysed for nanoluc luminescence or for single-cell suspension for flow cytometry, as described above, but with the HER2 Ab (R&D Systems, FAB9896V-100UG) used for flow cytometry applications.

### Data analysis
Statistical analyses of the data were performed using Prism 8.0 (GraphPad) by one- or two-way analysis of variance (ANOVA) for all $P$ values. All results are expressed as mean ± s.d. with upper and lower limits. Graphs were made in Prism 8.0 (GraphPad).

### Reporting summary
Further information on research design is available in the Nature Portfolio Reporting Summary linked to this article.

### Data availability
The raw and analysed datasets are available via figshare at https://doi.org/10.6084/m9.figshare.25338379 (ref. 49)

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

## Acknowledgements

O.P.B.W. discloses support for the research described in this study from the Swedish Research Council (VR) (2022-02449) and The Center for Innovative Medicine (CIMED) junior investigator grant (FoUI-976434). S.E.A. discloses support for the research described in this study from Horizon 2020 (EXPERT, 4-2298/2019), the Swedish Foundation of Strategic Research (FormulaEx, SM19-0007), European Research Council Consolidator Grant (DELIVER, 101001374), the Swedish Cancer Society (4-511/2022) and the Swedish Research Council (4–258/202). J.Z.N. discloses support for the research described in this study from the Swedish Research Council (VR) (2021-02407) and the CIMED junior investigator grant. A.G. is an International Society for Advancement of Cytometry Marylou Ingram Scholar 2019–2024 and supported by a Karolinska Institutet (KI) Network Medicine Alliance grant. Parts of the study were performed at the Live Cell imaging Core facility/Nikon Center of Excellence, at the KI, supported by grants from the Swedish Research Council, KI infrastructure and Centre for Innovative Medicine.

## Author contributions

O.P.B.W. conceived the idea. O.P.B.W., D.R.M. and S.E.A. designed the research. D.G., X.L. and D.R.M. designed the genetic constructs and performed the cloning. O.P.B.W., D.R.M., D.K.M., W.Z., R.J.W., T.S., A.M.Z., H.S., A.L., J.B., S.R., G.C., A.J.L. and A.G. performed the in vitro experiments. O.P.B.W., D.R.M. and A.J.L. performed the in vivo experiments. O.P.B.W. and D.R.M. analysed the results. R.J.W., A.M.Z., W.Z., M.A.-V., I.M., E.A., E.S., J.Z.N., D.G., A.G. and S.E.A. provided input on experimental plan and discussion of results. O.P.B.W. wrote the manuscript with input from all of the co-authors. O.P.B.W. and S.E.A. co-led the study.

## Funding

## Competing interests

S.E.A. is a founder of and a consultant for Evox Therapeutics. D.G., J.Z.N. and A.G. are consultants for Evox Therapeutics. O.P.B.W., D.G., J.Z.N., A.G. and S.E.A. have stock interest in Evox Therapeutics. H.S. and A.L. are employees of Evox Therapeutics. The other authors declare no competing interests.

## Additional information

**Correspondence and requests for materials** should be addressed to Oscar P. B. Wiklander or Samir EL Andaloussi.

# Reporting Summary

## Statistics

For all statistical analyses, confirm that the following items are present in the figure legend, table legend, main text, or Methods section.

| n/a | Confirmed | |
|---|---|---|
| ☐ | ☒ | The exact sample size (*n*) for each experimental group/condition, given as a discrete number and unit of measurement |
| ☐ | ☒ | A statement on whether measurements were taken from distinct samples or whether the same sample was measured repeatedly |
| ☐ | ☒ | The statistical test(s) used AND whether they are one- or two-sided *Only common tests should be described solely by name; describe more complex techniques in the Methods section.* |
| ☐ | ☒ | A description of all covariates tested |
| ☐ | ☒ | A description of any assumptions or corrections, such as tests of normality and adjustment for multiple comparisons |
| ☐ | ☒ | A full description of the statistical parameters including central tendency (e.g. means) or other basic estimates (e.g. regression coefficient) AND variation (e.g. standard deviation) or associated estimates of uncertainty (e.g. confidence intervals) |
| ☐ | ☒ | For null hypothesis testing, the test statistic (e.g. *F*, *t*, *r*) with confidence intervals, effect sizes, degrees of freedom and *P* value noted *Give P values as exact values whenever suitable.* |
| ☒ | ☐ | For Bayesian analysis, information on the choice of priors and Markov chain Monte Carlo settings |
| ☐ | ☒ | For hierarchical and complex designs, identification of the appropriate level for tests and full reporting of outcomes |
| ☐ | ☒ | Estimates of effect sizes (e.g. Cohen's *d*, Pearson's *r*), indicating how they were calculated |

*Our web collection on statistics for biologists contains articles on many of the points above.*

## Software and code

Policy information about availability of computer code

| Data collection | Microsoft Excel, NanoSight NTA software, MACSQuant Analyzer 10 flow cytometer, SoftMax Pro, Amnis ImageStream X Mk II and Amnis Cellstream software. Nanoimager S Mark II microscope software. |
|---|---|
| Data analysis | Prism 8-10, Microsoft excel, FlowJo. |

For manuscripts utilizing custom algorithms or software that are central to the research but not yet described in published literature, software must be made available to editors and reviewers. We strongly encourage code deposition in a community repository (e.g. GitHub). See the Nature Portfolio guidelines for submitting code & software for further information.

## Data

Policy information about availability of data

All manuscripts must include a data availability statement. This statement should provide the following information, where applicable:
- Accession codes, unique identifiers, or web links for publicly available datasets
- A description of any restrictions on data availability
- For clinical datasets or third party data, please ensure that the statement adheres to our policy

The raw and analysed datasets are available from figshare at https://doi.org/10.6084/m9.figshare.25338379.

## Human research participants

Policy information about studies involving human research participants and Sex and Gender in Research.

| | |
|---|---|
| Reporting on sex and gender | The study did not involve human participants. |
| Population characteristics | – |
| Recruitment | – |
| Ethics oversight | – |

Note that full information on the approval of the study protocol must also be provided in the manuscript.

# Field-specific reporting

Please select the one below that is the best fit for your research. If you are not sure, read the appropriate sections before making your selection.

☒ Life sciences ☐ Behavioural & social sciences ☐ Ecological, evolutionary & environmental sciences

For a reference copy of the document with all sections, see nature.com/documents/nr-reporting-summary-flat.pdf

# Life sciences study design

All studies must disclose on these points even when the disclosure is negative.

| | |
|---|---|
| Sample size | Sample sizes were determined without any pre-calculations of the number needed; instead, we used educated assumptions based on previous experience of the experimental setup of the in vitro and in vivo experiments. |
| Data exclusions | Data from one mouse at 24h in the Fc-EVs+PDL1-Ab was excluded as an outlier (which gave unreasonably high luminescent values, confirmed using the graphpads outlier calculator at https://www.graphpad.com/quickcalcs/grubbs1). |
| Replication | All attempts at replication were successful. |
| Randomization | Divisions into the different experimental groups were conducted in each experiment, after cells had been plated or mice had been inoculated, by randomly choosing wells or mice, respectively, for each treatment group. |
| Blinding | No blinding of the in vitro or in vivo work was conducted. All in vivo experiments were conducted together with at least two investigators, and the measurements were divided between the researchers on different days in to minimize bias. |

# Reporting for specific materials, systems and methods

We require information from authors about some types of materials, experimental systems and methods used in many studies. Here, indicate whether each material, system or method listed is relevant to your study. If you are not sure if a list item applies to your research, read the appropriate section before selecting a response.

## Materials & experimental systems

| n/a | Involved in the study |
|---|---|
| ☐ | ☒ Antibodies |
| ☐ | ☒ Eukaryotic cell lines |
| ☒ | ☐ Palaeontology and archaeology |
| ☐ | ☒ Animals and other organisms |
| ☒ | ☐ Clinical data |
| ☒ | ☐ Dual use research of concern |

## Methods

| n/a | Involved in the study |
|---|---|
| ☒ | ☐ ChIP-seq |
| ☐ | ☒ Flow cytometry |
| ☒ | ☐ MRI-based neuroimaging |

## Antibodies

| | |
|---|---|
| Antibodies used | rb-α-ms PD-L1 (APC, 485, Sino Biological), rt-α-ms CD3 ( 17A2, BioLegend), rt-α-ms CD8a (53-6.7, BioLegend), rt-α-ms CD45R/B220 (RA3.6B2, BioLegend), rt-α-ms CD11b (M1/70, BD Biosciences), rt-α-ms F4/80 (T45-2342, BD Biosciences), anti-CD63 (ab134045, Abcam), anti-Alix (MA1-83977, Thermofisher), Anti-NanoLuc (N7000, Promega), anti-Tsg101 (ab30871, Abcam), secondary antibody (Goat anti-Mouse (C00322), Goat anti-Rabbit (C90827-25), Rabbit anti-goat 10nm ab conjugated with gold nanoparticles (BBI |

Solutions), anti-CD9, Anti-CD63 and anti-CD81 detection antibodies (supplied in the MACSPlex Exosome Kit, human, Miltenyi Biotec), AlexaFluor647-conjugated human IgG Fc fragments (The Jackson Laboratory, cat 009-600-008), hIgG1-PE (Miltenyi; Cat. No. 130-113-438), hIgG4-PE (BioLegend; Cat. No. 403704), mIgG1-PE (Miltenyi; Cat. No. 130-113-200), and mIgG2-PE (Miltenyi; Cat. No. 130-092-215), Atezolizumab and Trastuzumab were kind gifts from Karolinsak University Hospital.

| Validation | The validation of the antibodies had been conducted by the distributor, and they were not further validated other than as described in the paper. |
|---|---|

## Eukaryotic cell lines

Policy information about cell lines and Sex and Gender in Research

| Cell line source(s) | FreeStyle 293F (HEK293FS; ThermoFisher Scientific), HEK293T (ATCC, CRL-3216), HeLa (ATCC, CCL-2), SKBR3 (ATCC, HTB-30), and B16F10 (ATCC, CRL-6475-LUC2). |
|---|---|
| Authentication | None of the cell lines were authenticated other then by morphology. |
| Mycoplasma contamination | All cell lines were confirmed negative by testing for mycoplasma. |
| Commonly misidentified lines (See ICLAC register) | No commonly misidentified cell lines were used. |

## Animals and other research organisms

Policy information about studies involving animals; ARRIVE guidelines recommended for reporting animal research, and Sex and Gender in Research

| Laboratory animals | C57BL/6 and Swiss nude mice. All mice were kept at a dedicated laboratory animal facility with controlled temperature, light-dark cycles, with access to nesting material, food and water. |
|---|---|
| Wild animals | The study did not involve wild animals. |
| Reporting on sex | Only female mice were used in order to reduce the number of mice and unwanted factors, owing to the fact that male mice may fight more with each other, especially when carrying diseases such as cancer. |
| Field-collected samples | The study did not involve samples collected from the field. |
| Ethics oversight | All of the animal experiments were performed in accordance with ethical permissions approved by The Swedish Local Board for Laboratory Animals, and designed to minimize the suffering and pain of the animals. |

Note that full information on the approval of the study protocol must also be provided in the manuscript.

## Flow Cytometry

### Plots

Confirm that:

☒ The axis labels state the marker and fluorochrome used (e.g. CD4-FITC).

☒ The axis scales are clearly visible. Include numbers along axes only for bottom left plot of group (a 'group' is an analysis of identical markers).

☒ All plots are contour plots with outliers or pseudocolor plots.

☒ A numerical value for number of cells or percentage (with statistics) is provided.

### Methodology

| Sample preparation | Sample preparations for either cells or EV measurements by flow cytometry are detailed in Methods. |
|---|---|
| Instrument | MACSQuant Analyzer 10 flow cytometer, Amnis ImageStream X Mk II, and Amnis Cellstream. |
| Software | The respective instrument's software was used to acquire the data. FlowJo was used for analysis. |
| Cell population abundance | Not applicable. |
| Gating strategy | The gating strategies are shown in the Supplementary Information. In short, for tumour-cell analysis, dead cells were excluded based on DAPI staining, and viable cells of interest were further analysed for their expression of mNG, CD3, CD8, B220, CD11b, F4/80, and/or PD-L1, respectively. For bead-based multiplex flow-cytometry analysis of EV surface markers (MACSPlex assay), single beads were identified based on FSC-H vs SSH-H and subsequent FSC-A/FSC-H gating, and fluorescent capture-bead populations were identified according to the manufacturer's recommendation and as described previously (Wiklander, Frontiers Immunol. 2018). For single-EV imaging flow cytometry, data were pre-gated on SSC(low) based on GFP- |

tagged biological reference material described extensively before (Görgens,JEV 2019).

☒ Tick this box to confirm that a figure exemplifying the gating strategy is provided in the Supplementary Information.

