## [Peer Review File · Nature Biomedical Engineering]

Antibody-displaying extracellular vesicles for targeted cancer therapy

Corresponding author: Samir EL Andaloussi

Editorial note

This document includes relevant written communications between the manuscript's corresponding author and the editor and reviewers of the manuscript during peer review. It includes decision letters relaying any editorial points and peer-review reports, and the authors' replies to these (under 'Rebuttal' headings). The editorial decisions are signed by the manuscript's handling editor, yet the editorial team and ultimately the journal's Chief Editor share responsibility for all decisions.

Any relevant documents attached to the decision letters are referred to as **Appendix #**, and can be found appended to this document. Any information deemed confidential has been redacted or removed. Earlier versions of the manuscript are not published, yet the originally submitted version may be available as a preprint. Because of editorial edits and changes during peer review, the published title of the paper and the title mentioned in below correspondence may differ.

Correspondence

Mon 24 Apr 2023

Decision on Article nBME-23-0703

Dear Prof EL Andaloussi,

Thank you again for submitting to *Nature Biomedical Engineering* your manuscript, "Targeted Therapy by Antibody Displaying Extracellular Vesicles". The manuscript has been seen by three experts, whose reports you will find at the end of this message. You will see that the reviewers appreciate the work, and that they raise a number of technical criticisms that we hope you will be able to address. In particular, we would expect that a revised version of the manuscript provides:

- * Quantification of the fraction of extracellular vesicles taken up by tumours, and of tumour-targeting specificity, as per the relevant queries of Reviewers #1 and #2.
- * Extended characterization of the binding affinities of IgG antibodies to Fc-binding extracellular vesicles.
- * Quantification of the levels of antibodies that can be conjugated onto the extracellular vesicles, and whether this depends on the type of antibody.
- * Discussion of the advantages of the approach with respect to existing strategies for the generation of extracellular vesicles that display specific antibodies.
- * Complete methodological details, as per the pertinent comments from all reviewers.

When you are ready to resubmit your manuscript, please upload the revised files, a point-by-point rebuttal to the comments from all reviewers, the reporting summary, and a cover letter that explains the main improvements included in the revision and responds to any points highlighted in this decision.

Please follow the following recommendations:- * Clearly highlight any amendments to the text and figures to help the reviewers and editors find and understand the changes (yet keep in mind that excessive marking can hinder readability).
- * If you and your co-authors disagree with a criticism, provide the arguments to the reviewer (optionally, indicate the relevant points in the cover letter).
- * If a criticism or suggestion is not addressed, please indicate so in the rebuttal to the reviewer comments and explain the reason(s).
- * Consider including responses to any criticisms raised by more than one reviewer at the beginning of the rebuttal, in a section addressed to all reviewers.
- * The rebuttal should include the reviewer comments in point-by-point format (please note that we provide all reviewers will the reports as they appear at the end of this message).
- * Provide the rebuttal to the reviewer comments and the cover letter as separate files.

We hope that you will be able to resubmit the manuscript within 16 weeks from the receipt of this message. If this is the case, you will be protected against potential scooping. Otherwise, we will be happy to consider a revised manuscript as long as the significance of the work is not compromised by work published elsewhere or accepted for publication at *Nature Biomedical Engineering*.

We hope that you will find the referee reports helpful when revising the work, which we look forward to receive. Please do not hesitate to contact me should you have any questions.

Best wishes,

Pep

Pep Pàmies
Chief Editor, Nature Biomedical Engineering

Reviewer #1 (Report for the authors (Required)):

In this manuscript, Andaloussi and colleagues have developed extracellular vesicles (EVs) that are equipped with Fc-binding domains, enabling the display of antibodies on their surface for targeted drug delivery or direct therapeutic effect. The authors have described the optimization procedure of the EVs and validated their efficacy as a potential platform for drug delivery. This work is promising as it provides a new approach for delivering antibodies or targeted drugs using EVs, which can generate broad interest in the field of drug delivery. However, following concerns should be addressed before potential acceptance.

1. Figure 2B requires clarification, as the gray column of TNFR is missing. Moreover, the authors should provide further explanation or discussion on why some proteins were expressed at high or low levels.
2. The authors should provide, at least, in vitro evidence of whether antibody-exchange happened and discuss the impact of such replacement on treatment efficacy.
3. There is an error in Figure 3F, where the upper line displays two identical images. This needs to be corrected.
4. The manuscript should quantify the amount of antibodies that can be conjugated onto EVs and explore or discuss whether the type of antibodies affects loading efficiency.
5. To validate the function of HER2-antibody in the process of HER2-mediated endocytosis, native HER2

antibodies should be used in combination with HER2-antibody-loaded EVs.

6. The authors should measure the distribution of EVs that reach tumors to assess their penetration depth, which is critical for anticancer activity. If the penetration capability is poor, the manuscript should consider discussing potential methods to improve it.

7. The pharmacokinetics of EVs before and after loading with antibodies need to be evaluated and compared.

Minor issues:

8. The authors should replace the old term "reticuloendothelial system (RES)" with "mononuclear phagocyte system (MPS)".

9. The authors should clarify the meaning of "uptake in tumor" and double-check the results or experiment procedure regarding T cells expressing PD-L1.

10. The authors should compare the antigen-binding capability of antibodies on EVs to that of free antibodies.

Reviewer #2 (Report for the authors (Required)):

In this study, the authors aimed to use extracellular vesicles (EVs) as nanocarriers for small molecule drug delivery, and to use antibodies to guide EVs to targeted tissues. For this purpose, the author designed to present a Fc-binding moiety on EVs (Fc-EVs) to enable antibody attachment on EVs. They screened pairs of 9 EV-membrane-sorting domains and 9 Fc-binding moieties, and eventually selected CD63 (as membrane-sorting domain) and Protein A's z domain (as the Fc-binding moiety) and made a fusion construct to produce Fc-EVs. Antibody-guided tumor-cell-targeting abilities of Fc-EVs were demonstrated using HER2 and PD-L1 antibodies. The authors further used an anti-PD-L1 antibody and B16F10 syngeneic mouse tumor model, and showed that anti-PDL1 antibody plus Fc-EVs achieved a better tumor-selective accumulation as compared to Ctrl Ab plus Fc-EVs. When the anti-PDL1/Fc-EVs were loaded with a small molecule drug, potent antitumor activity was achieved. This study presented an intriguing strategy for tumor-targeting therapy by combining tumor-specific antibodies and drug-loaded EVs.

However, some issues are not well explicated in this manuscript. One major issue is that Fc-EVs can bind with Fc receptors expressed on immune cells (e.g. B cells, macrophages, NK cells, etc.), which may lead to off-targeting toxicity of drug-loaded Fc-EVs. As shown in Fig.5E, accumulation of PD-L1 Ab + Fc-EVs was observed in lymph nodes, but the authors only explained this result from a PD-L1-dependent binding perspective. Other issues are listed as follows:

1. The difference and advantage of Fc-EVs in compare with reported antibody-guided synthetic nanoparticle delivery systems should be discussed.

2. Experiments should be conducted to investigate whether binding of Fc-binding domain with B cell receptors (BCR) on B cells or Fc receptors on other immune cells can lead to Fc-EV fusion with B cells or other immune cells.

3. The authors should use trastuzumab to study whether Ab+Fc-EVs could achieve tumor-selective accumulation, and Drug+Ab+Fc-binding EVs could achieve antitumor activity. Syngeneic tumor models constructed with B16F10 cells stably expressing human HER2 can be used. In such a model, HER2 expression is restricted to tumors, so that target-independent distribution of Fc-EVs can be assessed without interference of target-expressing normal tissue cells.

4. "the DNA sequences for all nine EV sorting domains, nine Fc-binders listed in Table 1", but no Table 1 can be found in this manuscript.

5. As shown in Fig. 2A, Fc-binding domains were inserted in EV-sorting domains. The authors should give more details of designs of EV-sorting domain and Fc-binding domain fusions.

6. Figure captions for Fig.2E is missing.
7. Binding affinity of different IgG isoforms (from both human and mouse) with Fc-EVs should be characterized. Binding affinity of anti-HER2 and anti-PD-L1 Abs used in this study with Fc-EVs should be characterized.
8. The author should give clear information of IgG used in Fig 3 including antibody name and antibody isoform.
9. "Next, to assess whether Fc-EVs would retain the displayed antibodies when exposed to blood, a competition assay was performed with mouse serum" Please explain the rationale of using mouse serum for competition but not human serum.
10. The authors should conduct experiments to observe dynamic changes of Fc-EVs in mouse plasma.
11. Fig. S4A indicates that different anti-PD-L1 Abs were used. Please clarify anti-PD-L1 Ab information in Fig. S4A and other related figures.
12. In Fig.5, the authors only presented tissue accumulation analysis 30 min post injection. The authors should analyze tissues accumulation of Fc-EVs at multiple timepoints after injection.
13. Please show number of mice in Fig.6.
14. Fig.6B and Fig.6C showed results of the same experiment? If mice are sacrificed or dead by Day 15 as shown in Fig. 6C, why the authors could record tumor volumes of all mice until Day 20 as shown in Fig.6B. Similar problem in Fig. 6E-F.
15. In Fig.6E-F, "Fc-Exo" is actually "Fc-EVs"? Why did not show mOS of the Fc-Exo+Dox+PDL1-Ab group in Fig. 6F?

Reviewer #3 (Report for the authors (Required)):

The submitted manuscript described development of a modular form of engineered extracellular vesicles (EVs) for antibody-mediated targeted therapy. This platform was generated by displaying antibody Fc domain-binding proteins on EV surface. Authors screened multiple EV sorting domains and antibody Fc-binding domains to identify ones with high expression on EVs using fluorescent cargo proteins and fluorescently labeled antibodies. Anti-HER2 antibodies and anti-PD-L1 antibodies were used as model targeting molecules for demonstrating antibody-mediated binding of the engineered EVs to HER2-positive and PD-L1 positive cells. Doxorubicin was loaded into such engineered EVs with associated anti-PD-L1 antibodies and evaluated for in vivo distribution, efficacy, and toxicity. Overall, the presented data suggest the selected antibodies could be associated with EVs through displayed Fc-binding domains to facilitate binding to target cells and uptake. In vivo studies showed accumulation in tumor cells and noticeable efficacy.

Despite these supporting results, the technical advancement and possible impact on targeted therapy resulting from this work are moderate, considering intrinsic weakness of this technique and established approaches for the same applications. While antibody Fc binding-domains could capture different types of antibodies for modular therapeutic development, this protein-protein interaction is non-covalent and reversible. Moreover, the binding affinity is moderate (>100 nM as measured in Fig. S3F), causing inadequate association of targeting antibody and continuous dissociation of captured antibodies in open systems. The targeting capability could not be sustained by such antibody-directed EVs. Quantification of bound antibodies on EV would be challenging due to dynamic association/dissociation, especially under non-equilibrium physiological conditions. In addition, displaying antibodies on EVs surface has been achieved in many different ways, such as directly fusing antibodies with the EV sorting domains, which ensure stable expression of antibodies without diffusion concerns. A recent paper (PMID: 36167305) also demonstrated display of antibodies on EVs by using glycosylphosphatidylinositol-anchored avidin and biotinylated antibodies. The tight binding affinity between avidin and biotin could ensure capture of targeting antibodies.

The reported EVs with associated PD-L1 antibodies were loaded with doxorubicin for targeted delivery. However, doxorubicin could be readily leaked from the EVs, causing potential toxicity. How to prevent release of such small-molecule drugs before reaching to tumors needs to be addressed.

A majority of quantitative analysis shown in this study was based on fluorescence signals, which may not be accurate as such signals are sensitive to local environment and molecular conformation, such as for quantification of loaded doxorubicin and EV-associated antibodies. More appropriate quantification methods should be used for the analytes.

Based on these major concerns, the submitted manuscript is not recommended for consideration for publication in this journal. In addition, the following issues need to be addressed.

1. Fig. 3B, Fc-EVs and mAb standards should be included as references for SEC analysis. In the lower panel, why was the mAb fluorescent signal not observed?
2. Fig. S3E, the samples of Fc-EV+Ab in PBS or serum should be compared with Fc-EV+Ab without incubation to evaluate possible antibody release. The released antibodies could be quantified by appropriate methods.
3. Fig. 4, to demonstrate the Fc-EVs+Ab could specifically bind to HER2- or PD-L1-positive cells, negative cell lines without HER2 or PD-L1 should be included.

Fri 15 Sep 2023

Decision on Article NBME-23-0703A

Dear Prof EL Andaloussi,

Thank you for your revised manuscript, "Targeted Therapy by Antibody Displaying Extracellular Vesicles", which has been seen by the original reviewers. In their reports, which you will find at the end of this message, you will see that the reviewers acknowledge the improvements to the work and raise a few additional technical criticisms, in particular about the pharmacokinetics of the extracellular vesicles, that I am hoping you will be able to satisfactorily address.

As before, when you are ready to resubmit your manuscript, please upload the revised files, a point-by-point rebuttal to the comments from all reviewers, the reporting summary, and a cover letter that explains the main improvements included in the revision and responds to any points highlighted in this decision.

As a reminder, please follow the following recommendations:

- * Clearly highlight any amendments to the text and figures to help the reviewers and editors find and understand the changes (yet keep in mind that excessive marking can hinder readability).
- * If you and your co-authors disagree with a criticism, provide the arguments to the reviewer (optionally, indicate the relevant points in the cover letter).
- * If a criticism or suggestion is not addressed, please indicate so in the rebuttal to the reviewer comments and explain the reason(s).
- * Consider including responses to any criticisms raised by more than one reviewer at the beginning of the rebuttal, in a section addressed to all reviewers.
- * The rebuttal should include the reviewer comments in point-by-point format (please note that we provide all reviewers will the reports as they appear at the end of this message).
- * Provide the rebuttal to the reviewer comments and the cover letter as separate files.

We look forward to receive a further revised version of the work. Please do not hesitate to contact me should you have any questions.

Best wishes,

Pep

Pep Pàmies
Chief Editor, Nature Biomedical Engineering

Reviewer #1 (Report for the authors (Required)):

The overall quality needs further significant enhancement. Several mentioned Figures in the response letter cannot be identified correctly in the manuscript. Also, the page number is missing, making the location of a specific page difficult. In addition to such sloppy issues, a few important issues raised by reviewers have not been well addressed.

Major issues:

1. In Figure S6E, I cannot find the results associated with the uptake of vesicles. In Figure S6H-J, I cannot identify the meaning of each column. Also, P values should be calculated.

2. The distribution or penetration of EV in the tumour has not been studied. Though an elevated level of accumulation of drugs in tumours has been identified, a detailed distribution of EV in the tumour can provide more information on the explanation of antitumour efficacy.

3. PK is critical for the anticancer activity of antibodies. The EV particles had a short circulation, which may substantially impact the loaded drug's anticancer activity. Thus, I suggest the authors add relevant discussion and the possible methods to improve this. More time points should be added to show the whole picture of PK. This is a basic property for such a new system.

Minor issues:

1. On page 4, the antibody replacement study was carried out for only one or ten minutes. This time scale is not sufficient to evaluate the IgG replacement in a real condition. Also, the fluorescence intensity should be quantified.

2. "APC" is widely used to denote antigen-presenting cells. To reduce possible misunderstanding by readers, I suggest the replacement of "APC" with another word. Also, what is the full spelling word for "APC"? It should appear when APC appears for the first time.

3. Scale bars are missing in Figure 3F.

4. The authors claimed updated Fig. S5F-G for studying the replacement of IgG. But I cannot find the data.

Reviewer #2 (Report for the authors (Required)):

One major issue on PK and two minor issues are as follows:

Major issue on PK : the authors did not address this comment, the authors should address this concern by extending the PK characterization to cover a longer time frame. The timepoints examined (30 minutes and 180 minutes) are too short to sufficiently capture the PK profile of biologics. Additionally, considering that the antitumor effect assessments were performed with a 3-day administration interval, it would be more appropriate to characterize the PK profile over a period of at least 3 days to better understand the drug's behavior in the body. This would provide a more comprehensive understanding of the PK properties and potentially correlate them with the observed antitumor effects.

For comment 13: Simply stating "n=10-20" is not precise enough. Please clarify the actual number of mice used in each group.

For comment 14: In the new Figure 7, it would be more reasonable to display the tumor volume growth curves of each mouse before they were sacrificed for better assessing the variability and overall trends in tumor size within the experimental groups.

Reviewer #3 (Report for the authors (Required)):

The revised manuscript includes a few additional experiments for addressing the raise concerns. However, the experimental designs such as using plasma in a closed system with short incubation for EVs would not address the disassociation of bound antibodies in living systems. As clearly shown in FigS9B, the loaded doxorubicin could be readily leaked from the EVs within 30 minutes, which would cause potential toxicity. Fluorescence-based quantitative analysis is not accurate for quantification of EV-associated molecules. In addition to these major technical issues, the possible impact and overall innovation of the reported approach on the targeted therapy are moderate, given different established methods in this area and the reversible binding nature between antibodies and EVs. It is therefore not recommended for acceptance of publication in this journal.

Tue 16 Jan 2024

Decision on Article NBME-23-0703B

Dear Prof EL Andaloussi,

Thank you for your patience in waiting for the feedback on your latest version of the revised manuscript, "Targeted Therapy by Antibody Displaying Extracellular Vesicles", which has been seen by the original reviewers. The reports are included at the end of this message.

You will see that Reviewer #2 is not satisfied with the additional pharmacokinetic data provided, and I hope that all the remaining technical points can be satisfactorily addressed. At the very minimum, discussion of how the pharmacokinetics of the engineered extracellular vesicles can be improved should be added. Also, please do discuss explicitly the related approach taken in this recent paper.

As before, when you are ready to resubmit your manuscript, please upload the revised files, a point-by-point rebuttal to the comments from Reviewers #2 and #3, the reporting summary, and a cover letter that explains the main improvements included in the revision and responds to any points highlighted in this decision.

Best wishes,

Pep

Pep Pàmies
Chief Editor, Nature Biomedical Engineering

Reviewer #1 (Report for the authors (Required)):

The authors have addressed my comments.

Reviewer #2 (Report for the authors (Required)):

1. The authors have not adequately addressed the issue of pharmacokinetics (PK). Though the authors conducted PK study up to 72 hours post injection, the PD-L1-Ab targeted Fc-EVs accumulation in tumor tissues was only numerically reported at 30 min post injection (Fig.5C). By interpreting the statistics data shown in Fig.5B, we can see that there were no significant differences between Fc-EV+PD-L1-Ab and control group (p values all >0.05) at time points beyond 3 hour. However, the authors missed to explicitly describe these critical data in the text in page 7, and missed to include a discussion around this.

2. Flaws in the PK data presentation in Fig.5 and Fig. S6:

- 1) Fig.5B seems to be repeatedly shown in Fig.S6 panel A.
- 2) No mentioning of time point examined for the data presented in Fig.S6C-J.
- 3) Fig.5 has panel A-I, but legend described additional panel J.
- 4) Fig. 5 legends for panel G-I are not consistent with the data presented in the figure.

3. For the tumor volume data in Fig 7, the suggestion was to present the tumor volume growth curves of the mice for each group before there was early sacrificed mice in the group, the early sacrificed mice in each group should not be included for calculation of the average tumor volume after they were sacrificed. In other words, the tumor volume growth curve for different group should not all end at the same time points. In the revision, the authors misunderstood the suggestion to present tumour size for the individual mice in each group, which resulted indistinguishable data of tumor growth trend –the same symbol used for every mice in one group at each timepoint (Fig7C and Fig7G).

Reviewer #3 (Report for the authors (Required)):

The overall innovation level is moderate, considering already published methods for antibody display (such as PMID: 36167305 and 37867228, which utilize glycosylphosphatidylinositol-anchored avidin and biotinylated antibodies and HaloTag fusion) as well as intrinsic weakness of this technique that depends on reversible non-covalent association. It is therefore not recommended for acceptance for publication in this journal.

Thu 13 Feb 2024

Decision on Article NBME-23-0703B

Dear Prof EL Andaloussi,

Thank you for your revised manuscript, "Targeted Therapy by Antibody Displaying Extracellular Vesicles". Having consulted with Reviewer #2 (whose brief comments you will find at the end of this message), I am pleased to write that we shall be happy to publish the manuscript in *Nature Biomedical Engineering*, provided that the points specified in the attached instructions file are addressed.

When you are ready to submit the final version of your manuscript, please upload the files specified in the instructions file.

We encourage authors to take up transparent peer review. If you are eligible and opt in to transparent peer review, we will publish, as a single supplementary file, all the reviewer comments for all the versions of the manuscript, your rebuttal letters, and the editorial decision letters. **If you opt in to transparent peer review, in the attached file please tick the box 'I wish to participate in transparent peer review'; if you prefer not to, please tick 'I do NOT wish to participate in transparent peer review'**. In the interest of confidentiality, we allow redactions to the rebuttal letters and to the reviewer comments. If you are concerned about the release of confidential data, please indicate what specific information you would like to have removed; we cannot incorporate redactions for any other reasons. More information on transparent peer review is available.

Best wishes,

Pep

Pep Pàmies
Chief Editor, Nature Biomedical Engineering

Reviewer #2 (Report for the authors (Required)):

The authors have addressed my comments. One minor note: In the response to comment #3, the correct figure reference should be Figure 7A-D not Fig. 5A-D.

Rebuttal 1

Thank you again for submitting to *Nature Biomedical Engineering* your manuscript, "Targeted Therapy by Antibody Displaying Extracellular Vesicles". The manuscript has been seen by three experts, whose reports you will find at the end of this message. You will see that the reviewers appreciate the work, and that they raise a number of technical criticisms that we hope you will be able to address.

We thank the editor for the positive comments and for bringing up very relevant questions. As you will see from the point-by-point replies below, we have performed extensive additional experiments to address the raised concerns. Among these include establishment of an additional breast cancer model that we used to validate FcEV targeting as well as employing a novel state of the art single particle analysis method (SPP) to enumerate antibody binding to EVs. By addressing all these questions, we believe that this has greatly improved the quality of the manuscript. Please find our answers in blue below to each remark.

In particular, we would expect that a revised version of the manuscript provides:

* Quantification of the fraction of extracellular vesicles taken up by tumours, and of tumour-targeting specificity, as per the relevant queries of Reviewers #1 and #2.

Thank you for addressing this relevant point. As the fraction of targeted EVs taken up by the tumours will depend on several factors, we have now added information on the number of EVs accumulated in the tumour tissue and what % of injected dose this corresponds to. Furthermore, expression of targeting molecule (PD-L1 in the B16F10 tumour mice model) in the dissected tumour tissue (based on single cell suspended tumour, assessed by flow cytometry) is now provided as well as the size of the tumours (Fig. S6C-D). In addition, we have now also included one more tumour mouse model, for which we have provided the same information (Fig. 6 and S8). The additional tumour model is based on inoculated hHER2+ breast cancer cells (SKBR-3) in swiss nude mice¹ for targeting with Fc-EVs displaying Trastuzumab (clinically used anti-HER2 Ab), to be consistent with the in vitro data (Fig. 4). Interestingly, both models showed a 3.5-fold increased Fc-EV tumour accumulation when displaying targeting Ab (Atezolizumab (PD-L1-Ab) in B16F10 and Trastuzumab (HER2) in SKBR-3 tumour bearing mice). In absolute numbers this corresponded to 6.3×10^8 EVs or 0.3% of the injected dose detected in the tumour at 30 min for Fc-EV+PD-L1-Ab in the B16F10 model (Fig S6A-B) and 4.6×10^5 (0.0002% of injected dose) of Fc-EVs in tumour of SKBR-3 bearing mice (Fig. S8A-B). The difference is explained by difference in tumour size (average 1200 mm^3 B16F10 tumours vs. average 60 mm^3 SKBR-3 tumours, Fig. S6C and S8C) and difference in expression of target protein (50% PDL1+ cells in B16F10 tumours, Fig S6D vs. 15% HER2+ cells in SKBR-3 tumours, Fig. S8D). Please see highlighted new sections in the manuscript (First paragraph p.8, last paragraph p.9) and updated Fig. 6, S6, and S8).

* Extended characterization of the binding affinities of IgG antibodies to Fc-binding extracellular vesicles.

This is indeed an important consideration and we have now characterized the binding affinities of IgG antibodies to Fc-EVs further. As expected based on the literature^{2,3}, the z-domain expressing Fc-EVs had the highest affinity for human IgG1 (hIgG1) Ab, which is highly relevant as most clinically approved Abs are in hIgG1-format. Please see added section in the manuscript (end of first paragraph p.4) and updated Fig. 3D showing the

affinity of two different human IgG subtypes and two different mouse IgG subtypes, with the affinity of different antibodies to Fc-EVs that follows the order: hIgG1 > hIgG4 > mIgG2 > mIgG1.

* Quantification of the levels of antibodies that can be conjugated onto the extracellular vesicles, and whether this depends on the type of antibody.

Thank you for emphasizing this relevant question. We have previously had issues to enumerate the number of Ab that can be displayed per EV in a detailed manner. By collaborating with other research colleagues (Dr. Taras Sych and Prof Erdinc Sezgin, who now are new co-authors), we were now able to utilize the newly developed single-particle profiler (SPP) method recently published in Nature Biotech ⁴. Using SPP we were able to determine that Fc-EVs display an average of 105 Ab per vesicle, whereas ctrl-EVs display ≤ 1 Ab/EV. Please see updated Fig. 3G-I & S3I-K and highlighted updated text in the end of p.4.

* Discussion of the advantages of the approach with respect to existing strategies for the generation of extracellular vesicles that display specific antibodies.

This is a highly relevant point, not only for this manuscript but also for the field of therapeutic EVs and nanomedicine in general. We have now expanded the last section of the manuscript with an “outlook” section including a discussion regarding this important topic. As there are several important considerations, parts of the discussion is also referred to relevant reviews discussing this topic.

* Complete methodological details, as per the pertinent comments from all reviewers.

Thank you for addressing this. Please see the updated method section with highlighted changes as well as the updated Fig. S12 and S13.

Reviewer #1 (Report for the authors (Required)):

In this manuscript, Andaloussi and colleagues have developed extracellular vesicles (EVs) that are equipped with Fc-binding domains, enabling the display of antibodies on their surface for targeted drug delivery or direct therapeutic effect. The authors have described the optimization procedure of the EVs and validated their efficacy as a potential platform for drug delivery. This work is promising as it provides a new approach for delivering antibodies or targeted drugs using EVs, which can generate broad interest in the field of drug delivery. However, following concerns should be addressed before potential acceptance.

We thank the reviewer for the positive comments, for thoroughly examining the manuscript and bringing up very relevant remarks. We have now addressed these remarks and we believe that this has greatly improved the quality and scientific strength of the manuscript. Please find our point-by-point replies below.

1. Figure 2B requires clarification, as the gray column of TNFR is missing. Moreover, the authors should provide further explanation or discussion on why some proteins were expressed at high or low levels.

Thank you for addressing this. The labelling with a fluorescent tracer (mNG) was aimed to be intraluminal and as TNFR only has the C-terminus intraluminally, there is only one line. This has now been clarified in the figure legend. The expression levels are discussed in the first paragraph of the result and discussion section. As this is merely part of the screen to optimize the Fc-binder on EVs, further explanation to the complexity of factors governing protein expression is outside the scope of this manuscript and restricted by the word limit.

2. The authors should provide, at least, in vitro evidence of whether antibody-exchange happened and discuss the impact of such replacement on treatment efficacy.

Thank you for raising this important concern. Please see highlighted added section in the manuscript (in the end of p.4) and updated Fig. S5F-G showing that no antibody exchange is taking place by additional incubation with different human or mouse IgG subtypes at the same concentrations (1:1-ratio of hIgG1 to other IgG subtypes), nor in human- or mouse plasma. We believe that this strengthens the in vivo data showing Ab-mediated Fc-EV targeting (which would not come about if there were strong antibody exchange). We have also added a discussion of regarding this with potential alternative approaches that could have been explored if antibody exchange would have been an issue including further functionalisation of the Ab display, as previously explored by others utilizing recombinant phosphatidylserine-binding of nanobodies⁵ or more recently by EV display of avidin that can be combined with biotinylated antibodies⁶.

3. There is an error in Figure 3F, where the upper line displays two identical images. This needs to be corrected.

Thank you very much for identifying this mistake. This has now been corrected in the updated Fig. 3F.

4. The manuscript should quantify the amount of antibodies that can be conjugated onto EVs and explore or discuss whether the type of antibodies affects loading efficiency.

We thank the reviewer for raising these relevant concerns. Regarding the number of antibodies that can be displayed on EVs, please see updated Fig. 3G-I & S3I-K and highlighted updated text in the end of p.4. Using the recently developed single-particle profiler method⁴ we were able to determine that Fc-EVs display an average of 105 Ab per vesicle, whereas ctrl-EVs display ≤ 1 Ab/EV. In regard to the testing of different antibody types, please refer to the added section in the manuscript (end of first paragraph p.4) and updated Fig. 3D showing the affinity of two different human IgG subtypes and two different mouse IgG subtypes, with the affinity of different antibodies to Fc-EVs that follows the order: hIgG1 > hIgG4 > mIgG2 > mIgG1. As mentioned above and in line with previous literature^{2,3}, the z-domain expressing Fc-EVs thus had the highest affinity for human IgG1 (hIgG1) Ab, which is highly relevant as most clinically approved Abs are hIgG1-based.

5. To validate the function of HER2-antibody in the process of HER2-mediated endocytosis, native HER2 antibodies should be used in combination with HER2-antibody-loaded EVs.

Thank you for raising this relevant point. Please find updated Fig. S6E and highlighted updated text in the second paragraph of p.6 in the revised manuscript. By pre-incubating cells with non-labelled antibody, uptake was significantly decreased, which confirms that the uptake is truly antibody driven. However, the decrease was quite modest, which is in line with the rapid recycling of HER2⁷.

6. The authors should measure the distribution of EVs that reach tumors to assess their penetration depth, which is critical for anticancer activity. If the penetration capability is poor, the manuscript should consider discussing potential methods to improve it.

Thank you for addressing this relevant topic. Please see highlighted new sections in the manuscript (First paragraph p.8, last paragraph p.9) and updated Fig. 6, S6, and S8). As mentioned above, the fraction of targeted EVs taken up by the tumours will depend on several factors and we have now added information on the number of EVs accumulated in the tumour tissue and what % of injected dose this corresponds to. Furthermore, expression of targeting molecule (PD-L1 for B16F10 tumour bearing mice) in the dissected tumour tissue (based on single cell suspended tumour assessed by flow cytometry) is now provided as well as the size of the tumors (Fig. S6C-D). In addition, we have now also included one more tumor mouse model, for which we have provided the same information (Fig. 6 and S8). The new tumour model is based on inoculation of hHER2+ breast cancer cells (SKBR-3) in swiss nude mice for targeting with Fc-EVs displaying Trastuzumab (clinically used anti-HER2 Ab) to be consistent with the in vitro data (Fig. 4). Interestingly, both models showed a 3.5-fold increased Fc-EV tumour accumulation when displaying targeting Ab for respective tumor target (Atezolizumab (PD-L1-Ab) in B16F10 and Trastuzumab (HER2-ab) in SKBR-3 tumour bearing mice). In absolute numbers this corresponded to 6.3×10^8 EVs or 0.3% of the injected dose detected in the tumour at 30 min for Fc-EV+PD-L1-Ab in the B16F10 model (Fig S6A-B) and 4.6×10^5 (0.0002% of injected dose) of Fc-EVs in tumour of SKBR-3 bearing mice (Fig. S8A-B). The difference is explained by difference in tumor size (average 1200 mm^3 B16F10 tumours vs. average 60 mm^3 SKBR-3 tumors, Fig. S6C and S8C) and difference in expression of target protein (50% PDL1+ cells in B16F10 tumours, Fig S6D vs. 15% HER2+ cells in SKBR-3 tumours, Fig. S8D).

7. The pharmacokinetics of EVs before and after loading with antibodies need to be evaluated and compared.

We thank the reviewer for the comment. We believe that larger pharmacokinetic (PK) studies are outside the scope of the current study, but based on previous PK studies on EVs^{8,9}, we conducted a comparison of EV accumulation in tissues and plasma at two different time points. In line with previous literature, the accumulation dropped substantially over time in all tissues and only tiny (<0.03% of input) levels of Fc-EVs were detectable in plasma at 180 minutes (Fig S7A-D). Interestingly, the tumour accumulation was still greater at 180 min for Fc-EVs guided by PD-L1-Ab than that of Fc-EVs with IgG-control at 30 min (Fig. S7A). Please see added section in the manuscript (last paragraph p.8-9) and Fig. S7 showing the pharmacokinetics of EVs before and after loading with antibodies at 30 min vs 180 min post injection.

Minor issues:

8. The authors should replace the old term "reticuloendothelial system (RES)" with "mononuclear phagocyte system (MPS)".

Thank you for addressing this, we have now updated it to the term "mononuclear phagocyte system (MPS)" as suggested.

9. The authors should clarify the meaning of "uptake in tumor" and double-check the results or experiment procedure regarding T cells expressing PD-L1.

Thank you for pointing this out. This has now been clarified in the figure legend (please see highlighted addition in the legend of Fig. 5, p.8). Regarding the interesting findings of PD-L1+ T-cells, which is coherent with previously published literature ¹⁰, this has been confirmed and is discussed in the third paragraph, p. 8.

10. The authors should compare the antigen-binding capability of antibodies on EVs to that of free antibodies.

We thank the reviewer for the comment. We have compared binding to PD-L1⁺ melanoma cells (B16F10) or HER2⁺ breast cancer cells (SKBR-3) with APC-labelled PD-L1- or HER-2-Ab respectively, with or without Fc-EVs. As seen in the figure below there are, as expected, no differences in Ab binding whether Fc-EVs are there or not (as measured by APC-positivity). We don't see how this data strengthens the manuscript and believe that it is excessive and has thus not included it in the manuscript.

Reviewer #2 (Report for the authors (Required)):

In this study, the authors aimed to use extracellular vesicles (EVs) as nanocarriers for small molecule drug delivery, and to use antibodies to guide EVs to targeted tissues. For this purpose, the author designed to present a Fc-binding moiety on EVs (Fc-EVs) to enable antibody attachment on EVs. They screened pairs of 9 EV-membrane-sorting domains and 9 Fc-binding moieties, and eventually selected CD63 (as membrane-sorting domain) and Protein A's z domain (as the Fc-binding moiety) and made a fusion construct to produce Fc-EVs. Antibody-guided tumor-cell-targeting abilities of Fc-EVs were demonstrated using HER2 and PD-L1 antibodies. The authors further used an anti-PD-L1 antibody and B16F10 syngeneic mouse tumor model, and showed that anti-PDL1 antibody plus Fc-EVs achieved a better tumor-selective accumulation as compared to Ctrl Ab plus Fc-EVs. When the anti-PDL1/Fc-EVs were loaded with a small molecule drug, potent antitumor activity was achieved. This study presented an intriguing strategy for tumor-targeting therapy by combining tumor-specific antibodies and drug-loaded EVs.

However, some issues are not well explicated in this manuscript. One major issue is that Fc-EVs can bind with Fc receptors expressed on immune cells (e.g. B cells, macrophages, NK cells, etc.), which may lead to off-targeting toxicity of drug-loaded Fc-EVs. As shown in Fig.5E, accumulation of PD-L1 Ab + Fc-EVs was observed in lymph nodes, but the authors only explained this result from a PD-L1-dependent binding perspective. Other issues are listed as follows:

We thank the reviewer for the positive comments, for thoroughly examining the manuscript and bringing up very relevant concerns. We have now addressed these remarks and we believe that this has greatly improved the scientific quality of the manuscript. Regarding the unnumbered issue of Fc-EVs binding Fc-receptors we would like to clarify that the Fc-EVs (as described in the 2nd and 3rd paragraph of p.2) don't display Fc-parts but an Fc-binding moiety, which minimizes this concern. Furthermore, if the uptake in lymph nodes would be governed by Fc-receptors independently of targeting Ab one would expect no difference between naked Fc-EVs or when Fc-EVs are displaying a PD-L1-Ab or IgG-control, which is contrasting our results which show significant differences (Fig. 5E).

1. The difference and advantage of Fc-EVs in compare with reported antibody-guided synthetic nanoparticle delivery systems should be discussed.

As mentioned above, this is a very relevant consideration, not only for this manuscript but also for the field of therapeutic EVs and nanomedicine in general. We have now expanded the last section of the manuscript with an "outlook" section including a discussion regarding this. As there are several important considerations parts of the discussion is also referred to relevant reviews discussing this topic.

2. Experiments should be conducted to investigate whether binding of Fc-binding domain with B cell receptors (BCR) on B cells or Fc receptors on other immune cells can lead to Fc-EV fusion with B cells or other immune cells.

We thank the reviewer for the comment. As explained above, it is unclear to us why an Fc-binder would bind another Fc-binder. However, we have compared the biodistribution of

Fc-EVs and ctrl-EVs (not included in the manuscript), which show no difference in distribution. If there were in fact an increased binding to immune cells, the distribution pattern should have been affected. Please see graph below:

3. The authors should use trastuzumab to study whether Ab+Fc-EVs could achieve tumor-selective accumulation, and Drug+Ab+Fc-binding EVs could achieve antitumor activity. Syngeneic tumor models constructed with B16F10 cells stably expressing human HER2 can be used. In such a model, HER2 expression is restricted to tumors, so that target-independent distribution of Fc-EVs can be assessed without interference of target-expressing normal tissue cells.

We thank the reviewer for the suggestion to include a HER2 tumor mouse model. To have a less synthetic system and to be coherent with the in vitro data (Fig. 4), we employed a xenograft mouse model bearing human HER2+ breast cancer by inoculating SKBR-3 cells in the mammary gland of mice¹. As mentioned above, a 3.5-fold increased Fc-EV tumour accumulation was seen when displaying the HER2-ab Trastuzumab, which is in line with the 3.5 fold increase seen for display of the PD-L1-Ab Atezolizumab in B16F10 tumours, thus strengthening the in vivo targeting findings. In absolute numbers this corresponded to 6.3×10^8 EVs or 0.3% of the injected dose detected in the tumour at 30 min for Fc-EV+PD-L1-Ab in the B16F10 model (Fig S6A-B) and 4.6×10^5 (0.0002% of injected dose) of Fc-EVs in tumour of SKBR-3 bearing mice (Fig. S8A-B). The difference is explained by difference in tumor size (average 1200 mm³ B16F10 tumours vs average 60 mm³ SKBR-3 tumors, Fig. S6C and S8C) and difference in expression of target protein (50% PDL1+ cells in B16F10 tumours, Fig S6D vs. 15% HER2+ cells in SKBR-3 tumours, Fig. S8D). In addition, in the syngeneic B16F10 tumour mice model, the distribution of Fc-EVs to spleen and liver were affected by display of Atezolizumab compared to IgG-ctrl (Fig. 5F-G). In contrast, and as expected, in a xenograft model with no human HER2+ cells present in normal tissues, no significant differences were seen in terms of spleen and liver distribution (Fig. 6D-E) for Fc-EV displaying trastuzumab or ctrl-IgG. This also further argue against the above raised concern regarding FcR mediated immune cell uptake. Please see highlighted new sections in the manuscript (First paragraph p.8, last paragraph p.9) and updated Fig. 6, S6, and S8).

4. “the DNA sequences for all nine EV sorting domains, nine Fc-binders listed in Table 1” , but no Table 1 can be found in this manuscript.

Thank you very much for identifying this mistake. This has now been updated in the method section and in the new Fig S13.

5. As shown in Fig. 2A, Fc-binding domains were inserted in EV-sorting domains. The authors should give more details of designs of EV-sorting domain and Fc-binding domain fusions.

Thank you for pointing this out. We have now expanded the information which can be found in the new Fig S12 and in the method section.

6. Figure captions for Fig.2E is missing.

Thank you for identifying this mistake, the figure legend has now been updated.

7. Binding affinity of different IgG isoforms (from both human and mouse) with Fc-EVs should be characterized. Binding affinity of anti-HER2 and anti-PD-L1 Abs used in this study with Fc-EVs should be characterized.

Thank you for emphasizing this this relevant point, which also was mentioned by one of the other reviewers. As stated above, binding affinity of different IgG types has now been assessed and is discussed in an added section of the manuscript (end of first paragraph p.4) and updated Fig. 3D. This shows the affinity of two different human IgG subtypes and two different mouse IgG subtypes to Fc-EVs that follows the order: hIgG1 > hIgG4 > mIgG2 > mIgG1. As mentioned above and in line with previous literature^{2,3}, the z-domain expressing Fc-EVs thus had the highest affinity for human IgG1 (hIgG1) Ab, which is highly relevant as most clinically approved Abs (including Trastuzumab and Atezolizumab) are in hIgG1-format with regard to the Fc-part.

8. The author should give clear information of IgG used in Fig 3 including antibody name and antibody isoform.

Thank you for addressing this, the figure legend has been updated to clarify this.

9. “Next, to assess whether Fc-EVs would retain the displayed antibodies when exposed to blood, a competition assay was performed with mouse serum” Please explain the rationale of using mouse serum for competition but not human serum.

We thank the reviewer for bringing up this concern, which is similar to points raised by the previous reviewer regarding antibody exchange. As mentioned above, to strengthen the potential of the Fc-EV platform, we have now remade this experiment to include subsequent exposure (to Fc-EV+hIgG1) of both mouse and human plasma, as well as different antibody types, at different time points. Please find highlighted added section in the manuscript (in the end of p.4) and updated Fig. S5F-G showing that no antibody exchange is taking place by additional incubation with different human or mouse IgG subtypes at the same concentrations (1:1-ratio of hIgG1 to other IgG subtypes), nor in human- or mouse plasma. We believe that this strengthens the in vivo data showing Ab-mediated Fc-EV targeting (which would not come about if there were strong antibody exchange).

10. The authors should conduct experiments to observe dynamic changes of Fc-EVs in mouse plasma.

Thank you for highlighting this concern. Please refer to the point above and the new section in the manuscript (in the end of p.4) as well as the updated Fig. S5F-G showing that no Fc-EV:antibody change is observed in human- or mouse plasma at early time points.

11. Fig. S4A indicates that different anti-PD-L1 Abs were used. Please clarify anti-PD-L1 Ab information in Fig. S4A and other related figures.

We thank the reviewer for bringing this to our attention. This has been clarified in the text (3rd paragraph p. 7) and figure legends (Fig. S4, 4, 5 & 7).

12. In Fig.5, the authors only presented tissue accumulation analysis 30 min post injection. The authors should analyze tissues accumulation of Fc-EVs at multiple timepoints after injection.

Thank you for addressing this relevant point. As mentioned above, we argue that larger pharmacokinetic (PK) studies are outside the scope of the current study, but based on previous PK studies on EVs^{8,9}, we conducted a comparison of EV accumulation in tissues and plasma at two different time points. In line with previous literature, the accumulation dropped substantially over time in all tissues and only tiny (<0.03% of input) levels of Fc-EVs were detectable in plasma at 180 minutes (Fig S7A-D). Interestingly, the tumour accumulation was still greater at 180 min for Fc-EVs guided by PD-L1-Ab than that of Fc-EVs with IgG-control at 30 min (Fig. S7A). Please see added section in the manuscript (last paragraph p.8-9) and Fig. S7 showing the pharmacokinetics of Fc-EVs before and after loading with antibodies at 30 min vs 180 min post injection.

13. Please show number of mice in Fig.6.

We thank the reviewer for the comments. As described in the figure legend (of Fig. 7, which previously was Fig. 6) the number of mice at the start is 10-20 in these experiments. We have now clarified this further in the legend in parallel to your next concern, please see below.

14. Fig.6B and Fig.6C showed results of the same experiment? If mice are sacrificed or dead by Day 15 as shown in Fig. 6C, why the authors could record tumor volumes of all mice until Day 20 as shown in Fig.6B. Similar problem in Fig. 6E-F.

We thank the reviewer for pointing this out. The recorded tumor volumes are of the mice that are still alive. This explains why the error bars are larger in the control groups in the end. Of note, if the mice would not have been sacrificed due to human endpoint, the differences of the groups (in favor of the Fc-EVs+Dox+PD-L1-Ab group) would have been even greater. This has now have been clarified in the figure legend, Fig. 7 (Fig. 6 in previous manuscript version).

15. In Fig.6E-F, "Fc-Exo" is actually "Fc-EVs"? Why did not show mOS of the Fc-Exo+Dox+PDL1-Ab group in Fig. 6F?

We thank the reviewer for pointing out this mistake, which now have been corrected in Fig. 7F (former Fig. 6F)

Reviewer #3 (Report for the authors (Required)):

The submitted manuscript described development of a modular form of engineered extracellular vesicles (EVs) for antibody-mediated targeted therapy. This platform was generated by displaying antibody Fc domain-binding proteins on EV surface. Authors screened multiple EV sorting domains and antibody Fc-binding domains to identify ones with high expression on EVs using fluorescent cargo proteins and fluorescently labeled antibodies. Anti-HER2 antibodies and anti-PD-L1 antibodies were used as model targeting molecules for demonstrating antibody-mediated binding of the engineered EVs to HER2-positive and PD-L1 positive cells. Doxorubicin was loaded into such engineered EVs with associated anti-PD-L1 antibodies and evaluated for in vivo distribution, efficacy, and toxicity. Overall, the presented data suggest the selected antibodies could be associated with EVs through displayed Fc-binding domains to facilitate binding to target cells and uptake. In vivo studies showed accumulation in tumor cells and noticeable efficacy.

Despite these supporting results, the technical advancement and possible impact on targeted therapy resulting from this work are moderate, considering intrinsic weakness of this technique and established approaches for the same applications. While antibody Fc binding-domains could capture different types of antibodies for modular therapeutic development, this protein-protein interaction is non-covalent and reversible. Moreover, the binding affinity is moderate (>100 nM as measured in Fig. S3F), causing inadequate association of targeting antibody and continuous dissociation of captured antibodies in open systems. The targeting capability could not be sustained by such antibody-directed EVs. Quantification of bound antibodies on EV would be challenging due to dynamic association/dissociation, especially under non-equilibrium physiological conditions. In addition, displaying antibodies on EVs surface has been achieved in many different ways, such as directly fusing antibodies with the EV sorting domains, which ensure stable expression of antibodies without diffusion concerns. A recent paper (PMID: 36167305) also demonstrated display of antibodies on EVs by using glycosylphosphatidylinositol-anchored avidin and biotinylated antibodies. The tight binding affinity between avidin and biotin could ensure capture of targeting antibodies.

We thank the reviewer for bring up the concern regarding affinity and antibody exchange. Owing to these concerns we have now remade the affinity measurements, including that of different IgG subtypes from mice and human (Fig. 3D). As mentioned above and as hypothesized based on the literature^{2, 3}, the z-domain expressing Fc-EVs had the highest affinity for human IgG1 (hIgG1) Ab, which is highly relevant as most clinically approved Abs are in hIgG1-format. Please see added section in the manuscript (end of first paragraph p.4) and updated Fig. 3D showing the affinity of two different human IgG subtypes and two different mouse IgG subtypes, with the affinity of different antibodies to Fc-EVs that follows the order: hIgG1 > hIgG4 > mIgG2 > mIgG1. Furthermore, we have also added a section in the manuscript (in the end of p.4) and introduced new data, updated Fig. S5F-G, showing that no antibody exchange is taking place by additional incubation with different human or mouse IgG subtypes at the same concentrations (1:1-ratio of hIgG1 to other IgG subtypes), nor in human- or mouse plasma. We believe that this strengthens the in vivo data showing Ab-mediated Fc-EV targeting (which would not come about if there were strong antibody

exchange). We have also added a discussion of regarding this with potential alternative approaches that could have been explored if antibody exchange would have been an issue including further functionalisation of the Ab display, as previously explored by others utilizing recombinant phosphatidylserine-binding of nanobodies⁵ or more recently by EV display of avidin that can be combined with biotinylated antibodies⁶.

The reported EVs with associated PD-L1 antibodies were loaded with doxorubicin for targeted delivery. However, doxorubicin could be readily leaked from the EVs, causing potential toxicity. How to prevent release of such small-molecule drugs before reaching to tumors needs to be addressed.

We thank the reviewer for raising this important point. To address this concern with electroporation causing permeabilization of the membrane, the stability of encapsulation of Dox was assessed in plasma at 37 degrees (Fig S9B). Ensuringly, no significant decrease could be detected at 15 minutes, even if a slight decrease of 20% was observed at 30 min. Given that EVs rapidly distribute to their target tissue within minutes^{8,9}, and the relatively low dose of Dox required when encapsulated in EVs¹¹, PD-L1 targeted Dox loaded Fc-EVs were considered a highly attractive candidate for proof-of-concept use of the Fc-EV platform in cancer settings. This has now also been included in the text (please see highlighted section in the end of the first paragraph p.10).

A majority of quantitative analysis shown in this study was based on fluorescence signals, which may not be accurate as such signals are sensitive to local environment and molecular conformation, such as for quantification of loaded doxorubicin and EV-associated antibodies. More appropriate quantification methods should be used for the analytes. We thank the reviewer for the comment. The technical tools used for quantitative analysis herein are indeed based on fluorescence and luminescence that (as all technical measurements) have limitations, which also is discussed in the first paragraph of p.4. However, all experiments are thus conducted with appropriate controls and with complementary analyses when appropriate and technically feasible. As mentioned above, changes in e.g. Dox were detectable using the current set-up. Of note, we have included several modern state-of-the-art methods, which have been developed based on small differences in fluorescence signals including IFC¹², Nanoimager¹³, and SPP⁴, which all emphasize that, while being aware of potential limitations, fluorescent based methods are not only valid, but also of great value and extremely important in nanomedicine research.

Based on these major concerns, the submitted manuscript is not recommended for consideration for publication in this journal. In addition, the following issues need to be addressed.

1. Fig. 3B, Fc-EVs and mAb standards should be included as references for SEC analysis. In the lower panel, why was the mAb fluorescent signal not observed?

We thank the reviewer for spotting this error, the figure has now been updated.

2. Fig. S3E, the samples of Fc-EV+Ab in PBS or serum should be compared with Fc-EV+Ab without incubation to evaluate possible antibody release. The released antibodies could be quantified by appropriate methods.

We thank the reviewer for the advice. A related set-up has now been added (in accordance with suggestion by the other reviewers and the editor). As mentioned above, we have now added a section in the manuscript (in the end of p.4) and introduced new data, updated Fig. S5F-G, showing that no antibody exchange is taking place by additional incubation with different human or mouse IgG subtypes at the same concentrations (1:1-ratio of hIgG1 to other IgG subtypes), nor in human- or mouse plasma.

3. Fig. 4, to demonstrate the Fc-EVs+Ab could specifically bind to HER2- or PD-L1-positive cells, negative cell lines without HER2 or PD-L1 should be included.

We thank the reviewer for the recommendation and have now included negative cell lines as suggested. As hypothesized, display of PD-L1-Ab on Fc EVs had no effect on uptake in PD-L1 negative SKBR-3 cells and HER2-Ab had no effect in HER2⁻ B16F10 cells (Fig. S4C-D), which further underpins the target specificity of Fc-EVs. We have also added this information in the updated manuscript (see highlighted text in the first paragraph of p.8).

References:

1. Rodallec, A. et al. Tumor uptake and associated greater efficacy of anti-Her2 immunoliposome does not rely on Her2 expression status: study of a docetaxel-trastuzumab immunoliposome on Her2+ breast cancer model (SKBR3). *Anticancer Drugs* **31**, 463-472 (2020).
2. Langone, J.J., Boyle, M.D. & Borsos, T. Studies on the interaction between protein A and immunoglobulin G. II. Composition and activity of complexes formed between protein A and IgG. *J Immunol* **121**, 333-338 (1978).
3. Nilsson, B. et al. A synthetic IgG-binding domain based on staphylococcal protein A. *Protein Eng* **1**, 107-113 (1987).
4. Sych, T. et al. High-throughput measurement of the content and properties of nano-sized bioparticles with single-particle profiler. *Nat Biotechnol* (2023).
5. Kooijmans, S.A.A., Gitz-Francois, J., Schiffelers, R.M. & Vader, P. Recombinant phosphatidylserine-binding nanobodies for targeting of extracellular vesicles to tumor cells: a plug-and-play approach. *Nanoscale* **10**, 2413-2426 (2018).
6. Sabani, B. et al. A novel surface functionalization platform to prime extracellular vesicles for targeted therapy and diagnostic imaging. *Nanomedicine* **47**, 102607 (2023).
7. Cheng, J. et al. Molecular Mechanism of HER2 Rapid Internalization and Redirected Trafficking Induced by Anti-HER2 Biparatopic Antibody. *Antibodies (Basel)* **9** (2020).
8. Gupta, D. et al. Quantification of extracellular vesicles in vitro and in vivo using sensitive bioluminescence imaging. *J Extracell Vesicles* **9**, 1800222 (2020).
9. Walker, S. et al. Extracellular vesicle-based drug delivery systems for cancer treatment. *Theranostics* **9**, 8001-8017 (2019).
10. Diskin, B. et al. PD-L1 engagement on T cells promotes self-tolerance and suppression of neighboring macrophages and effector T cells in cancer. *Nat Immunol* **21**, 442-454 (2020).
11. Lennaard, A.J., Mamand, D.R., Wiklander, R.J., El Andaloussi, S. & Wiklander, O.P.B. Optimised Electroporation for Loading of Extracellular Vesicles with Doxorubicin. *Pharmaceutics* **14** (2021).

12. Gorgens, A. et al. Optimisation of imaging flow cytometry for the analysis of single extracellular vesicles by using fluorescence-tagged vesicles as biological reference material. *J Extracell Vesicles* **8**, 1587567 (2019).
13. McNamara, R.P. et al. Imaging of surface microdomains on individual extracellular vesicles in 3-D. *J Extracell Vesicles* **11**, e12191 (2022).

Rebuttal 2

Thank you for your revised manuscript, "Targeted Therapy by Antibody Displaying Extracellular Vesicles", which has been seen by the original reviewers. In their reports, which you will find at the end of this message, you will see that the reviewers acknowledge the improvements to the work and raise a few additional technical criticisms, in particular about the pharmacokinetics of the extracellular vesicles, that I am hoping you will be able to satisfactorily address.

We thank the editor and the reviewers for the positive comments and for bringing up very relevant questions. As you will see from the point-by-point replies below, we have performed extensive additional experiments to address the raised concerns. Among these include pharmacokinetics study up to 72 hours post injection and extended antibody dissociation experiments. By addressing all these questions, we believe that this has greatly improved the quality of the manuscript. Please find our answers in blue below each remark and with turquoise highlighted changes in the manuscript.

Reviewer #1 (Report for the authors (Required)):

The overall quality needs further significant enhancement. Several mentioned Figures in the response letter cannot be identified correctly in the manuscript. Also, the page number is missing, making the location of a specific page difficult. In addition to such sloppy issues, a few important issues raised by reviewers have not been well addressed.

We thank the reviewer for thoroughly examining the manuscript and bringing up very relevant remarks. We have now addressed these remarks and we believe that this has greatly improved the quality and scientific strength of the manuscript. Please find our point-by-point replies below. We sincerely apologise for the typos. We have now updated the text accordingly and also added page numbers.

Major issues:

1. In Figure S6E, I cannot find the results associated with the uptake of vesicles. In Figure S6H-J, I cannot identify the meaning of each column. Also, P values should be calculated.

Thank you for identifying this mistake in the figure legend, which has now been updated. Fig. S6E-F shows the percentage of T-cells (CD3+) and CD8-neg population of these T-cells, respectively. We have also clarified that the p-values were non-significant if not indicated.

2. The distribution or penetration of EV in the tumour has not been studied. Though an elevated level of accumulation of drugs in tumours has been identified, a detailed distribution of EV in the tumour can provide more information on the explanation of antitumour efficacy.

We thank the reviewer for the comment. We here clearly demonstrate in vitro (Fig. 4) and in vivo targeting (Fig. 5 & 6) with two different targets (HER2 and PD-L1) with increased accumulation in tumour tissue, as well as functional in vivo data with clear antitumoral efficacy (Fig. 7). Though interesting, we believe that the intratumoral distribution is outside the scope of the manuscript and could be further researched in subsequent studies. With that said, with the intent to address this remark, tumour tissue has been processed for immunohistochemistry in parallel with the newly conducted extensive (about 150 mice) in vivo PK and biodistribution experiment (see point below). However, due to technical staining issues, we have not been able to produce this data within a reasonable time frame.

3. PK is critical for the anticancer activity of antibodies. The EV particles had a short circulation, which may substantially impact the loaded drug's anticancer activity. Thus, I suggest the authors add relevant discussion and the possible methods to improve this. More time points should be added to show the whole picture of PK. This is a basic property for such a new system.

Thank you for addressing this. This was also raised by the other reviewers. We have now conducted a pharmacokinetics study up to 72 hours post injection which highlights sustained increased accumulation in tumour tissue of PD-L1-Ab targeted Fc-EVs (compare to naked or IgG-ctrl displaying Fc-EVs) up to 72h, whereas the EV accumulation in spleen, liver and plasma drops quickly. We further see that Fc-EVs with PD-L1-Ab have a slight reduction in spleen and liver accumulation. The half-life in blood is as expected short, with $t_{1/2}$ of 3-4 minutes. Please see updated Fig. 5 and S6 as well as highlighted updated text p. 7. Collectively, we believe that this additional data significantly strengthens the manuscript and thank the reviewers for the input.

Minor issues:

1. On page 4, the antibody replacement study was carried out for only one or ten minutes. This time scale is not sufficient to evaluate the IgG replacement in a real condition. Also, the fluorescence intensity should be quantified.

We have now extended the experiment to include a 30-minute timepoint as well, which we believe is more than enough considering the short half-life of EVs in blood (as discussed above). We have now also indicated the mean fluorescence intensity (MFI), please see updated Fig s3F-G and updated text on p. 4.

2. "APC" is widely used to denote antigen-presenting cells. To reduce possible misunderstanding by readers, I suggest the replacement of "APC" with another word. Also, what is the full spelling word for "APC"? It should appear when APC appears for the first time.

We thank the reviewer for indicating the need for clarification. We understand that APC is an abbreviation for multiple terms. In addition to antigen-presenting cells it is also commonly used for the fluorophore APC (Allophycocyanin) and we can thus not exchange the wording for this, but have clarified this on p. 2.

3. Scale bars are missing in Figure 3F.

We have now updated the figure to include a scale bar (please see Fig. 3F and updated figure legend). Thank you for spotting this.

4. The authors claimed updated Fig. S5F-G for studying the replacement of IgG. But I cannot find the data.

We apologise for the typo and refer the reviewer to Fig S3F-G, which has been updated accordingly as discussed under the first point of minor issues above.

Reviewer #2 (Report for the authors (Required)):

One major issue on PK and two minor issues are as follows:

Major issue on PK : the authors did not address this comment, the authors should address this concern by extending the PK characterization to cover a longer time frame. The timepoints examined (30 minutes and 180 minutes) are too short to sufficiently capture the PK profile of biologics. Additionally, considering that the antitumor effect assessments were performed with a 3-day administration interval, it would be more appropriate to characterize the PK profile over a

period of at least 3 days to better understand the drug's behavior in the body. This would provide a more comprehensive understanding of the PK properties and potentially correlate them with the observed antitumor effects.

We thank the reviewer for emphasizing this point. As discussed above we have followed this recommendation and now conducted a pharmacokinetics study up to 72 hours post injection which highlights sustained increased accumulation in tumour tissue of PD-L1-Ab targeted Fc-EVs (compared to naked or IgG-ctrl displaying Fc-EVs) up to 72h, whereas the EV accumulation in spleen, liver and plasma drops quickly. We further see that Fc-EVs with PD-L1-Ab have a slight reduction in spleen and liver accumulation. The half-life in blood is as expected short, with $T_{1/2}$ of 3-4 minutes. Please see updated Fig. 5 and S6 as well as highlighted updated text p. 7. This suits well with the 3-day dosing interval and we believe that this additional data further strengthens the manuscript and thank the reviewers for the input.

For comment 13: Simply stating "n=10-20" is not precise enough. Please clarify the actual number of mice used in each group.

We thank the reviewer for indicating the need for clarification. We have now updated the figure to show n number per group, please see updated Fig. 7 and corresponding figure legend.

For comment 14: In the new Figure 7, it would be more reasonable to display the tumor volume growth curves of each mouse before they were sacrificed for better assessing the variability and overall trends in tumor size within the experimental groups.

We have now updated Fig. 7 to also include tumour size for the individual mice in each group over time as suggested as this gives a better insight into the variability within the groups. We believe it is important to still keep the average growth curves to assess the differences between the treatment groups. Please see Fig. 7 and corresponding figure legend.

Reviewer #3 (Report for the authors (Required)):

The revised manuscript includes a few additional experiments for addressing the raise concerns. However, the experimental designs such as using plasma in a closed system with short incubation for EVs would not address the disassociation of bound antibodies in living systems.

We thank the reviewer for the remark. As discussed above, we have now extended the experiment to include a 30-minute timepoint as well, which we believe is more than enough considering the short half-life of EVs in blood (as discussed above, as well as in updated text p.8 and shown in Fig. 5D). Please see updated Fig s3F-G and updated text p. 4. In addition to these in vitro experiments, we further believe that the clear and significant targeting effects seen in the in vivo experiments following systemic injections (p. 7-9, including Fig. 5 & 6) proves that the potential disassociation of bound antibodies is not a crucial issue here.

As clearly shown in FigS9B, the loaded doxorubicin could be readily leaked from the EVs within 30 minutes, which would cause potential toxicity.

We thank the reviewer for the comment. As mentioned end of p. 9 in the manuscript, a slight decrease of 20% was observed at 30 min for doxorubicin. Considering the reduced doses of doxorubicin used and the uptake of Fc-EVs being the highest at 30 min, we consider that this is a

clear benefit compared to the risk of naked doxorubicin. Furthermore, the toxicity assessment using these doses could not detect any toxicity (Fig. S10A-D and discussed on p. 11 of the manuscript).

Fluorescence-based quantitative analysis is not accurate for quantification of EV-associated molecules. In addition to these major technical issues, the possible impact and overall innovation of the reported approach on the targeted therapy are moderate, given different established methods in this area and the reversible binding nature between antibodies and EVs. It is therefore not recommended for acceptance of publication in this journal.

As mentioned in our previous response, the technical tools used for quantitative analysis herein are indeed based on fluorescence and luminescence that (as all technical measurements) have limitations, which also is pointed out in the first paragraph of p.4. All experiments are thus conducted with appropriate controls and with complementary analyses when appropriate and technically feasible. As mentioned above, changes in e.g. Dox were detectable using the current set-up. Of note, we have included several modern state-of-the-art methods, which have been developed based on small differences in fluorescence signals including IFC¹, Nanoimager², and SPP³, which all emphasize that, while being aware of potential limitations, fluorescent based methods are not only valid, but also of great value and extremely important in nanomedicine research.

References:

1. Gorgens, A. et al. Optimisation of imaging flow cytometry for the analysis of single extracellular vesicles by using fluorescence-tagged vesicles as biological reference material. *J Extracell Vesicles* **8**, 1587567 (2019).
2. McNamara, R.P. et al. Imaging of surface microdomains on individual extracellular vesicles in 3-D. *J Extracell Vesicles* **11**, e12191 (2022).
3. Sych, T. et al. High-throughput measurement of the content and properties of nano-sized bioparticles with single-particle profiler. *Nat Biotechnol* (2023).

Rebuttal 3

We thank the editor and the reviewers for the comments and relevant questions. Please see the point-by-point replies below addressing the raised concerns. Please find our answers in blue below each remark and highlighted changes in turquoise in the updated manuscript.

Reviewer #1 (Report for the authors (Required)):

The authors have addressed my comments.

We thank the reviewer for all the valuable remarks in the previous manuscript versions and we are happy that these now have been satisfactorily addressed.

Reviewer #2 (Report for the authors (Required)):

1. The authors have not adequately addressed the issue of pharmacokinetics (PK). Though the authors conducted PK study up to 72 hours post injection, the PD-L1-Ab targeted Fc-EVs accumulation in tumor tissues was only numerically reported at 30 min post injection (Fig.5C). By interpreting the statistics data shown in Fig.5B, we can see that there were no significant differences between Fc-EV+PD-L1-Ab and control group (p values all >0.05) at time points beyond 3 hour. However, the authors missed to explicitly describe these critical data in the text in page 7, and missed to include a discussion around this.

Thank you for addressing this. For the PK data, the accumulation at each timepoint is indicated by the y-axis of Fig. 5B. To clarify the difference between the groups at each time point, similar to that of Fig. 5C, we have now updated Fig. S6 to include the fold difference at each timepoint (except 30 min, which is already shown in Fig. 5C), please see new Fig. S6A-F. In addition, when assessing the differences between the groups at each timepoint (i.e. one-way ANOVA), there is a significant increase of EVs in the tumour for the Fc-EV+PD-L1-Ab group at each time point (from 15 min to 72h) in both updated Fig 5B and S6A-G. In the previous version of the manuscript the p-values had accidentally been derived from an incorrect two-way ANOVA, which was noted when reviewing the data set once again. We have now updated Fig 5 and Fig S6 accordingly and updated the text, see highlighted section on p7. We hope this clarifies the issue

2. Flaws in the PK data presentation in Fig.5 and Fig. S6:

This has now been clarified in the highlighted section on p7, updated Fig. 5 & S6, as mentioned above.

1) Fig.5B seems to be repeatedly shown in Fig.S6 panel A.

The data is from the same experiment, but as described in the text (and in the figures), Fig. 5B shows tumour accumulation by EVs per gram tissue, whereas Fig. S6G shows percentage of injected EV dose accumulating in tumour.

2) No mentioning of time point examined for the data presented in Fig.S6C-J.

Thank you for spotting this. To clarify that previous Fig. S6C-J is examined at 30 min post injection, we have now made this into a separate Figure (new Fig. S7) and updated the figure legend.

3) Fig.5 has panel A-I, but legend described additional panel J.

Thank you for seeing this typo. We have now updated the figure legend of Fig. 5.

4) Fig. 5 legends for panel G-I are not consistent with the data presented in the figure.

Thank you for seeing this typo (originating from the same as above). We have now updated the figure legend of Fig. 5.

3. For the tumor volume data in Fig 7, the suggestion was to present the tumor volume growth curves of the mice for each group before there was early sacrificed mice in the group, the early sacrificed mice in each group should not be included for calculation of the average tumor volume after they were sacrificed. In other words, the tumor volume growth curve for different group should not all end at the same time points. In the revision, the authors misunderstood the suggestion to present tumour size for the individual mice in each group, which resulted indistinguishable data of tumor growth trend –the same symbol used for every mice in one group at each timepoint (Fig7C and Fig7G).

To us it is still unclear what the reviewer is asking for. The tumour volume of sacrificed mice is of course not included in the calculation of the tumour volume for that group at the time points after they have been sacrificed. Since there are remaining mice in all treatment groups at the endpoint of the experiment running until day 20 (Fig. 5A-D) as shown in the survival curve Fig. 5D, the tumour volume curves in Fig. 5B all end at day 20. This contrasts with the longer experiment (Fig. 5E-H) where no mice in the PBS group remain after day 20, and no mice in the Fc-EV+Dox+IgG-ctrl group after day 26, compared to 80% remaining at the endpoint of day 35 in the Fc-EV+Dox+PDL1-Ab group. Here (Fig. 7F), the tumour volume curves thus end at different time points. In your previous comment you stated “In the new Figure 7, it would be more reasonable to display the tumor volume growth curves of each mouse before they were sacrificed for better assessing the variability and overall trends in tumor size within the experimental groups.”, which is why we showed a detailed graph with the tumour size for each individual mouse and assigned them a symbol to indicate to which group they belong. This we believe gives a better insight into the variability within each group. If you meant to suggest that the tumour volume curve of each mouse should be displayed this would make the data indistinguishable as the data set includes 80 mice and thus 80 lines.

Reviewer #3 (Report for the authors (Required)):

The overall innovation level is moderate, considering already published methods for antibody display (such as PMID: 36167305 and 37867228, which utilize glycosylphosphatidylinositol-anchored avidin and biotinylated antibodies and HaloTag fusion) as well as intrinsic weakness of this technique that depends on reversible non-covalent association. It is therefore not recommended for acceptance for publication in this journal.

As mentioned in the previous answers to the reviewer, the article with PMID 36167305 is already discussed in the manuscript (last section p.4). We have also previously addressed

your concerns regarding “reversible non-covalent association” with extensive significant data and discussion; please refer to the previous rebuttals. We thank the reviewer for pointing out the even more recent study with PMID 37867228, which was submitted to and accepted in the journal months after this manuscript was initially submitted. Both articles show rather limited functional data and lack any in vivo targeting. The articles do however strengthen that the concept of displaying antibodies on the surface of EVs is feasible (which we also have patented since 2020, US20200109183A1). In this manuscript we optimize the platform, extensively characterize the Fc-EV platform, and show that this novel concept can be utilized for targeting in vitro and in vivo. In addition, we include a functional readout in vivo as we take the strategy yet further and show therapeutic efficacy in a cancer in vivo model. In addition, in the most recent study (PMID 37867228), antibody display is just one out of several applications that the authors use to show their engineering strategy. This part entails one figure and half a page of written text. The characterization is very limited and there is no information on plasma stability, number of Abs per EV, etc. The targeting shown is also very modest, 1.4-fold increased uptake of targeted EVs compared to control EVs in vitro, which is limited to only one cell target in vitro. In the other publication (PMID 36167305) suggested by the reviewer the difference in uptake is not even calculated, but merely indicated by IHC images. We here show efficient targeting to two different cell targets with fold increase of 339-fold to HER2+ cells and 509-fold to PD-L1+ cells in vitro and also show significant targeting of the same cellular targets in vivo. We have now also included a discussion regarding these previous publications which can be found highlighted on p. 11, in addition to the above- mentioned discussion on p. 4 (last paragraph).